# Worldwide Statistical Correlation of Eight Years of *Swarm* Satellite Data with M5.5+ Earthquakes: New Hints about the Preseismic Phenomena from Space

Dedalo Marchetti [1], Angelo De Santis [2], Saioa A. Campuzano [3], Kaiguang Zhu [1,*], Maurizio Soldani [2], Serena D'Arcangelo [2,4], Martina Orlando [2], Ting Wang [1], Gianfranco Cianchini [2], Domenico Di Mauro [2], Alessandro Ippolito [2], Adriano Nardi [2], Dario Sabbagh [2], Wenqi Chen [1], Xiaodan He [1], Xuhui Shen [5], Jiami Wen [1], Donghua Zhang [1], Hanshuo Zhang [1], Yiqun Zhang [1] and Zhima Zeren [5]

1. College of Instrumentation and Electrical Engineering, Jilin University, Changchun 130061, China; dedalomarchetti@jlu.edu.cn (D.M.); tingwang20@mails.jlu.edu.cn (T.W.); wqchen21@mails.jlu.edu.cn (W.C.); hexd19@mails.jlu.edu.cn (X.H.); wenjm20@mails.jlu.edu.cn (J.W.); zhangdh6518@mails.jlu.edu.cn (D.Z.); zhanghs6518@mails.jlu.edu.cn (H.Z.); yiqun21@mails.jlu.edu.cn (Y.Z.)
2. Istituto Nazionale di Geofisica e Vulcanologia, 00143 Rome, Italy; angelo.desantis@ingv.it (A.D.S.); maurizio.soldani@ingv.it (M.S.); serena.darcangelo@ingv.it (S.D.); martina.orlando@ingv.it (M.O.); gianfranco.cianchini@ingv.it (G.C.); domenico.dimauro@ingv.it (D.D.M.); alessandro.ippolito@ingv.it (A.I.); adriano.nardi@ingv.it (A.N.); dario.sabbagh@ingv.it (D.S.)
3. Instituto de Geociencias IGEO (CSIC-UCM), 28040 Madrid, Spain; saioa.arquero@igeo.ucm-csic.es
4. Facultad de Ciencias Físicas, Universidad Complutense de Madrid (UCM), 28040 Madrid, Spain
5. Space Observation Research Center, National Institute of Natural Hazards, MEMC, Beijing 100085, China; xuhuishen@ninhm.ac.cn (X.S.); zerenzhima@ninhm.ac.cn (Z.Z.)
* Correspondence: zhukaiguang@jlu.edu.cn

**Abstract:** Nowadays, the possibility that medium-large earthquakes could produce some electromagnetic ionospheric disturbances during their preparatory phase is controversial in the scientific community. Some previous works using satellite data from DEMETER, *Swarm* and, recently, CSES provided several pieces of evidence supporting the existence of such precursory phenomena in terms of single case studies and statical analyses. In this work, we applied a Worldwide Statistical Correlation approach to **M5.5+ shallow earthquakes** using the first **8 years of *Swarm*** (i.e., from November 2013 to November 2021) **magnetic field and electron density signals** in order to improve the significance of previous statistical studies and provide some new results on how earthquake features could influence ionospheric electromagnetic disturbances. We implemented new methodologies based on the hypothesis that the anticipation time of anomalies of larger earthquakes is usually longer than that of anomalies of smaller magnitude. We also considered the **signal's frequency** to introduce a new identification criterion for the anomalies. We find that taking into account the frequency can improve the statistical significance (up to 25% for magnetic data and up to 100% for electron density). Furthermore, we noted that the **frequency** of the *Swarm* **magnetic field signal** of possible precursor anomalies seems to slightly **increase as the earthquake is approaching**. Finally, we checked a possible relationship between the frequency of the detected anomalies and earthquake features. The earthquake focal mechanism seems to have a low or null influence on the frequency of the detected anomalies, while the epicenter location appears to play an important role. In fact, **land** earthquakes are more likely to be preceded by **slower** (lower frequency) magnetic field signals, whereas **sea** seismic events show a higher probability of being preceded by **faster** (higher frequency) magnetic field signals.

**Keywords:** electromagnetic; frequency; *Swarm*; earthquake; precursors; ionosphere

## 1. Introduction

In this paper, we will deal with the delicate research on the possible precursors of earthquakes, i.e., those anomalous phenomena that might appear before earthquakes because they are caused by their preparation phase and thus anticipate their imminent occurrence. The earthquake precursors could be simply classified as seismic precursors and non-seismic precursors: the former are eventual modifications of seismicity or seismological parameters before the occurrence of an earthquake, while the latter, the object of our present research, are alterations of some geophysical parameters, such as atmospheric chemical and physical characteristics, radon emission, and, furthermore, possible electromagnetic disturbances up to the ionosphere [1–3]. Historically, some evidence of electromagnetic precursors has been reported in the data from ground magnetic observatories—for example, before the M6.9 Spitak (Armenia) 1988 and M7.1 Loma-Prieta 1989 earthquakes by Fraser-Smith et al. [4] and Molchanov et al. [5]. A model to explain the generation of Ultra Low Frequency (ULF) electromagnetic waves responsible for such pre-earthquake disturbance was proposed by Molchanov and Hayakawa [6] and based on electromagnetic pulses produced by the separation of electric charges at the fault level due to micro-fracturing activity during the preparation phase of the earthquake. Variations of this explanation have been proposed; e.g., [7], introducing a better model for microfracturing rate variation on the time and improving the geometry of the fault. However, one of the most important theories was put forward by Freund [8,9], which proposed the generation of positive holes (called p-holes) due to the peroxy defects on the rocks also induced by stress on the rock. Freund proposed such theory as a "universal" explanation of all other precursors: for example, he proposed that the detected Thermal-InfraRed anomalies could be explained by photon emission in such wavelengths due to the recombination at the Earth's surface of the peroxy bonds, as well as their location atop topographical profiles by the electrical migration of p-holes in the lithosphere due to the "tip electric effect" [9]. Even if p-holes are a different source with respect to the one initially proposed, the propagation mechanism of the eventual pre-earthquake ULF waves could, in principle, remain unchanged. Other theories have also been developed and warrant consideration—for example, supposing that the release of p-holes at the Earth's surface can produce an accumulation of electric charges that could gradually create a perturbation in the atmosphere and then propagate up to the ionosphere in the form of plasma bubble, interacting and following the magnetic field, as numerically simulated by Kuo et al. [10,11]. Such simulation opened a scientific discussion, as Prokhorov and Zolotov [12] found that some assumptions in the model were too simplified, and Kuo and Lee replied with an update of the original work, showing that the model seems, in any case, valid [13]. Furthermore, Denisenko et al. [14,15] found some possible inaccuracies in the model of Kuo et al., proposing another model that improved the calculus of the conductivity in the ionosphere, but the value on the ground of the electric field strength signal necessary to create a small perturbation of the ionosphere was found to be very large, even if it is still possible. As stated by the same authors of all these papers, more pieces of evidence are necessary to understand if such phenomena exist and which could be the most reliable model to describe them. This paper aims to provide some further observational points for such discussion.

According to Pulinets and Ouzounov [16], the ionization of the air could be also produced by the release of radon instead of (or in addition to) the p-holes. Additionally, other mechanisms for electric (or electromagnetic) disturbances have been proposed based on the change of the resistivity of rocks caused by the movements of fluids (the highly conductive water that fills the new cracks) in the crust before the occurrence of the earthquake under the so-called "Dilatancy Model" by Scholz et al. [17]. For instance, the movement of fluids in Central Italy prior to the occurrence of significant earthquakes has been detected from seismological analyses [18]. Changes in resistivity have been supposed and calculated to describe the atmospheric electric and magnetic field anomalies detected by a ground antenna [19]. Furthermore, ground and atmospheric observations, together with *Swarm* magnetic field satellite anomalous data, have been proposed as good candi-

dates to describe a pre-earthquake chain of anomalies [20]. A comprehensive review of several models of the Lithosphere–Atmosphere–Ionosphere (or, in some cases, just the Lithosphere–Ionosphere) Coupling models has been provided by Liperovsky et al. [21], where other possible physical couplings are also proposed such as pre-earthquake acoustic gravity waves, atmospheric electric currents possibly generated by the release of charged aerosol, variations in the D and E layers in the ionosphere induced by radon release, a direct resonator between the lithosphere and the ionosphere that is supposed to form a spherical capacitor in this model, and even more challenging possible explanations. Previous statistical works have provided pieces of evidence of the correlation of the ionospheric disturbances in terms of electron density measured by the DEMETER satellite, finding a significant increase from 10 to 6 days prior to the seismic M4.8+ events [22–24], and an increase in the electric field (typically around 10/20 mV/m) from a few minutes to some days prior to several M5+ earthquakes was reported by Zolotov [25] using DEMETER and INTERCOSMOS-BULGARIA-1300 satellites. Previous studies by some of the authors of this paper analyzed the *Swarm* and China Seismo Electromagnetic Satellite (CSES) data prior to the occurrence of about 20 earthquakes in recent years with a magnitude "M" in the range $6.0 \leq M \leq 8.3$ [26–28]. The worldwide systematic association of detected ionospheric disturbances in the magnetic field and electron density with earthquakes has been statistically proved by De Santis et al. [29], and a preliminary investigation of CSES-01 electron density data has also shown promising results [30].

The present work is the natural extension of the first statistical and systematic analysis of correlation using satellite magnetic field data from *Swarm* satellites with global M5.5+ shallow earthquakes [29]. Here, a temporal extension of the analysis is provided in terms of analyzed time (i.e., 8 years instead of 4.7 years) and the preparation time before the earthquake (from 1000 days before, instead of 500 days before). Furthermore, some new methodologies are also provided to corroborate the results.

In Section 2, we provide a brief introduction about the investigated dataset and the data analysis method. Section 3 illustrates the results, which are deeply discussed in Section 4. In Section 5, we present some conclusions and future perspectives.

## 2. Materials and Methods

### 2.1. Investigated Datasets

This study investigates satellite datasets from the *Swarm* satellite mission and earthquakes from the United States Geological Survey (USGS) catalog, as described in the following sections.

### 2.1.1. Satellite Datasets

In this paper, we analyze the ionospheric disturbances retrieved from satellite in situ measurements possibly related to earthquakes. In particular, we analyze data from the *Swarm* three-identical satellite constellation launched by the European Space Agency (ESA) on 22 November 2013. At the present time (April 2022), all three satellites are still in orbit, with most instruments in good condition. The *Swarm* constellation is the "state of the art" to monitor with the best accuracy the spatial structure and the temporal evolution of the Earth's magnetic field from space [31]. All three *Swarm* satellites include identical payloads onboard. In particular, in this work, the Y-East component of the magnetic field and electron density (*Ne*) are analyzed as originally measured by fluxgate magnetometers (Vector Field Magnetometer, VFM) and Langmuir probes (Electric Field Instrument, EFI), respectively. The Y-East magnetic field component has been selected, as it is expected to be more sensitive to the internal perturbations [32] and, in the case of Lithosphere–Atmosphere–Ionosphere Coupling (LAIC) coupling, to the Field Aligned Currents (FAC) induced by air ionization from radon release or p-holes. At the ionospheric heights, these currents are expected to be almost horizontal, while at the ground/atmospheric level they are expected to be almost vertical [21]. Among the two X-North and Y-East horizontal components, the Y-East component should be more sensitive to such kinds of perturbations, as it is quasi-orthogonal

to the main geomagnetic field generated by the Earth's outer core, so we expect that the same absolute perturbation is more evident on the Y-East component. The used magnetic field data are at "Low-Resolution" and are provided at a sample rate of 1 Hz in a daily file that contains 86,400 samples associated with 31 or 32 tracks (i.e., half orbits) per day. *Swarm Ne* is sampled at 2 Hz and is provided in the Electric Field Instrument (EFI) product. There is an open debate on the discrepancies in the *Ne* absolute value calibration measured with another satellite in a similar orbit, the China Seismo-Electromagnetic Satellite (CSES). For instance, it is known that the absolute value of electron density measured by CSES-01 (i.e., the first satellite of the CSES series) is lower than the ones estimated by *Swarm* [33] and by the IRI-2016 ionospheric model (e.g., comparison in the Indonesia area in [34]). However, such an absolute difference (bias) does not affect our results because our algorithms work on residuals, i.e., they are based on variations measured along the track, which are well consistent between the two satellite missions [33,34].

This paper analyzes the nighttime and daytime tracks for both the magnetic field (B) and electron density (*Ne*) datasets. In fact, as the investigation approach is track-by-track, we do not expect that the solar-driven daytime activity can affect our result in terms of "false anomalies", even if some potential pre-earthquake anomalies could be lost inside the higher day noise (especially for *Ne*).

### 2.1.2. Earthquake Catalog Acquisition and Pre-Processing

We retrieved the M5+ global earthquakes (EQs) that occurred in the same analyzed time interval from the USGS global catalog, which provides an estimation of the hypocenter, origin time, and magnitude (generally, the moment magnitude Mw calculated from the centroid moment tensor inversion, or, in alternative, the short-period body wave magnitude mb or other values). We then applied a declustering technique using Reasenberg's [35] method to remove aftershocks and foreshocks. With respect to the previous work of De Santis et al. [29], here, a slightly larger time window is utilized—10 days before and 20 days after the events—to search for other possible related events of the same cluster, while the other parameters were unchanged (i.e., a radius of 10 km, magnitude cut-off of 5.0, confidence level of 95%, hypocenter uncertainty according to the catalog). For further details on the declustering of the catalog and how this impacts this work, the reader is invited to see Appendix A. In the declustered catalog, only the shallow (depth $\leq$ 50 km) EQs with a magnitude greater than or equal to 5.5 are selected as the seismic dataset for further processing. In the first 8 years of *Swarm*, slightly fewer than 2200 shallow M5.5+ earthquakes were finally selected.

### 2.2. Methods of Analysis

Firstly, the *Swarm* satellite data are pre-processed by MAgnetic *Swarm* anomaly detection by Spline analysis (MASS) or electron density (*Ne*) Anomaly Detection (NeAD) algorithms to search for anomalies. The approach applied to the magnetic field and electron density data is basically the same, and only some parameters are settled differently to consider the different sample frequencies and signal-to-noise ratios properly. MASS and NeAD perform the following operations while taking into account the data along the track between $-50°$ and $+50°$ geomagnetic latitude (see Figure S1 in the Supplementary Materials as an example of data processing):

1.  First differences of the data: the difference between two consecutive data divided by the time interval between them (as a first order approximation of the temporal derivative).
2.  Removal of the residual trend: a cubic spline coming from the fitting of the derivative of the data (as obtained from Step 1) is removed.
3.  Analysis of the residuals obtained from Step 1 and Step 2 along the track: moving windows of 7° latitude are used, with an incremental shift of 1/5 of the window length (i.e., ~1.4° of latitude).

4.   The root mean square (rms) is calculated inside the window and compared with the whole track's Root Mean Square (RMS). The anomaly is defined if $rms > k_t \cdot RMS$ ($k_t$ is chosen as 2.5 or 3 if no-frequency investigation is performed; more details are given later on).

In addition, a new frequency analysis is carried out, and the Fast Fourier Transform (FFT) is calculated for each moving window and for the whole spectrum. Three sub frequency bands are divided according to their period ranges: (2–10) s, (10–25) s, and (25–50) s. In each frequency band, the anomaly is defined if the mean intensity overpasses the mean *FFT* intensity plus $K_{FFT}$ times its standard deviation σ (for example, for the 2–10 s band, the anomaly is defined if $FFT(2s \le p \le 10s) > FFT + K_{FFT} \cdot \sigma(FFT)$. This criterion on $K_{FFT}$ is in addition to Step 4 about $k_t$, but it is not applied for the basic anomaly extraction (i.e., when no frequency analysis is made). We recognize that the residuals with respect to the spectral mean are not Gaussian, but it is feasible to introduce the standard deviation simply as an operational measure of the data dispersion around the mean of the spectrum. Even if other quantities (e.g., the absolute mean deviation or the largest deviation) could be used, we found that the results are practically the same, so we preferred to use the standard deviation that can be easily and quickly estimated.

The two datasets (satellite anomalies and earthquakes) are investigated by the Worldwide Statistical Correlation (WSC) algorithm fully described in [29] that is only briefly summarized here.

The WSC algorithm first extracts the subset of anomalies and earthquakes to be correlated by a Superposed Epoch and Space Approach (SESA). The anomalies are selected by applying several thresholds. In particular, we only select tracks in geomagnetic quiet time (i.e., when the geomagnetic indices satisfy the following conditions: $|\text{Dst}| \le 20$ nT and $a_p \le 10$ nT), with all samples acquired in nominal satellite conditions (checked by the "Quality Flags" of the *Swarm* satellite, according to the respective operating manuals), and put conditions over the rms of the window and, for some analyses, an additional threshold on the frequency. For each earthquake of magnitude *M*, the anomalies inside its preparation area, as defined by a circular area with a Dobrovolsky's radius (in km: $10^{(0.43 \cdot M)}$ [36]) that occurred in the analyzed window (for example, from 500 days before the earthquake until 500 days after it), are extracted. If an anomaly is univocally associated with one earthquake, this couple (anomaly–earthquake) is defined. In case the anomaly is compatible with more than one seismic event, one of the following four methods is used to select which earthquake is associated with the anomaly:

1.   **Method 1 "All anomalies–EQs"**. This method selects all earthquakes compatible with the investigated anomaly. The advantage is that we do not apply any assumption for the analysis, and we can suppose that, in case the statistics are sufficient, the "wrong" anomaly–earthquake couples increase just the background without creating additional artificial concentrations of anomalies. On the other side, one disturbance can be produced only by one earthquake, and so, unless the signal is superposed (or under the satellite resolution), the anomaly could not be associated with more than one event. For this reason, we also introduce the following methods.

2.   **Method 2 "Min [log(ΔT R)]"**. This method selects the closest earthquake in space ("R") and time (anticipation time $\Delta T = time_{EQ} - time_{anomaly}$). The selection is made by searching for the minimum of the following equation: $\log_{10}(|\Delta T \cdot R|)$. The criterion is based on the assumption that an anomaly is more likely to be produced by the closest earthquake in space and time.

3.   **Method 3 "Max (magnitude)"**. This method selects the earthquake with the highest magnitude in the space and time domains of interest. The assumption is that a larger earthquake produces more anomalies before its occurrence that can be detectable in the ionosphere.

4. **Method 4 "Closer (Rikitake)"**. This method takes into account that a larger earthquake with a magnitude $M$ is expected to have a longer anticipation time $\Delta T$ of its possible precursors, according to the Rikitake law [37], expressed as:

$$\Delta T = a + b \cdot M \tag{1}$$

With this method, the anticipation time $\Delta T$ is calculated using the $a$ and $b$ coefficients found for the *Swarm* satellite data (magnetic field or *Ne*) by De Santis et al. [29] and inserted in the Rikitake law expression. On the other side, the decimal logarithm of the real anticipation time is calculated, and the earthquake that presents the minimum difference with respect to the Rikitake law is selected.

Please note that the first three methods are the same as those used by De Santis et al. [29], while here, a new fourth method is also proposed. Hereafter, if no indication is given about the method, the analysis will follow Method 1, i.e., all anomalies are associated with all the compatible earthquakes. Finally, it is worth noticing that any method presents advantages and disadvantages, as we do not know the exact mechanism behind the generation of ionospheric disturbances (even if several theories have been proposed). We can rely on the concentrations of anomalies confirmed by several of these methods and expect that a future better understanding of the LAIC mechanism can improve the selection of the best of the above methods or help formulate a new one. Of course, all methods take advantage of the fact that all analyses are retrospective. It would be more difficult to adapt the methods for a real time analysis rather than establish a significant correlation between ionospheric anomalies and earthquakes, but this is not within the present scope of the work.

The results are illustrated in a "standard" representation graph (see, for example, Figure 1), where the horizontal axis represents the time of the anomaly with respect to the earthquake occurrence (i.e., negative days before the earthquake occurrence and positive days after), and the vertical axis is the distance of the detected anomaly in terms of degrees from the epicenter. Therefore, each anomaly is placed on this graph in terms of distance in time and space from its associated earthquake (according to one of the four previously listed methods). All the anomalies and earthquakes are superposed on the same graph, and, in the end, the results are binned in 50 horizontal and 10 vertical cells (or bins). For each bin, the number of anomalies falling in the bin is divided by its geographic area (circle or annulus), obtaining the anomalies' density represented by a color bar scale. For the first row (which represents the 50 temporal bins closer to the epicenters), the maximum concentration of such density is searched, and its location (pxmax), the number of anomalies (Nmax), the number of EQs associated with these anomalies (NEQmax), and the total sum of the released energy ($E_{EQ}$) is reported in the second line of the heading. Furthermore, for such a maximum density/concentration bin, other statistical quantities are evaluated and described in the following paragraph.

To evaluate if the results obtained from the WSC of real earthquake and anomaly data are statistically significant, 30 random space-time homogeneous distributions of anomalies, with the same number of anomalies as each real data analysis, have been generated. Then, the WSC algorithm has been applied with the same parameters used for the real satellite anomalies. In particular, for the maximum concentration, we estimate the quantity, $D_{MAX}/D_0$, that represents the density of anomalies in the maximum ($D_{MAX}$) with respect to a theoretical homogeneous distribution of anomalies ($D_0$):

$$D_{MAX}/D_0 = \frac{N_{anom}^{MAX} \cdot \Delta T^{Tot} \cdot Area^{Tot}}{N_{anom}^{Tot} \cdot \Delta T^{pixel} \cdot Area^{pixel} \cdot n_{EQ}} \tag{2}$$

where $N_{anom}^{MAX}$ is the number of anomalies in the maximum pixel (which, in the title of the graphs, is indicated as "$N_{MAX}$") that has a duration $\Delta T^{pixel}$ and covers an area $Area^{pixel}$. The total number of identified anomalies (independently from the earthquakes) is $N_{anom}^{Tot}$, and it is also indicated in the first line of the heading of the graph as the second number (63,178 in the case of Figure 1). The whole analyzed area (i.e., the spherical zone between

50°S and 50°N of geomagnetic latitude) is $Area^{Tot}$, and the total analyzed time is $\Delta T^{Tot}$ (in this work, it is 8 years). The quantity $D_{MAX}/D_0$ has been calculated for real analysis, i.e., $[D_{MAX}/D_0]_{real}$, and its mean random value $\overline{[D_{MAX}/D_0]}_{random}$ and its standard deviation (σ) have been estimated over a set of 30 random simulations. By these quantities, we defined the two statistical parameters, *d* and *n*:

$$d = \frac{[D_{MAX}/D_0]_{real}}{[D_{MAX}/D_0]_{random}} ; n = \frac{[D_{MAX}/D_0]_{real} - \overline{[D_{MAX}/D_0]}_{random}}{\sigma([D_{MAX}/D_0]_{random})} \tag{3}$$

According to the above definitions, *d* represents how many times the real maximum concentration is higher than the one expected from random simulations, and *n* estimates how many standard deviations (of random simulations) the real concentration is higher than the random one. The number of random simulations was chosen as a function of the results in terms of the reached stability of the obtained values of *d* and *n*. We found that, generally, even with one or two random simulations, the estimation of *d* is accurate up to one decimal digit (if the number of anomalies is greater than 30,000), while *n* requires more random runs, i.e., at least 15 or 30 simulations for about 30,000 anomalies.

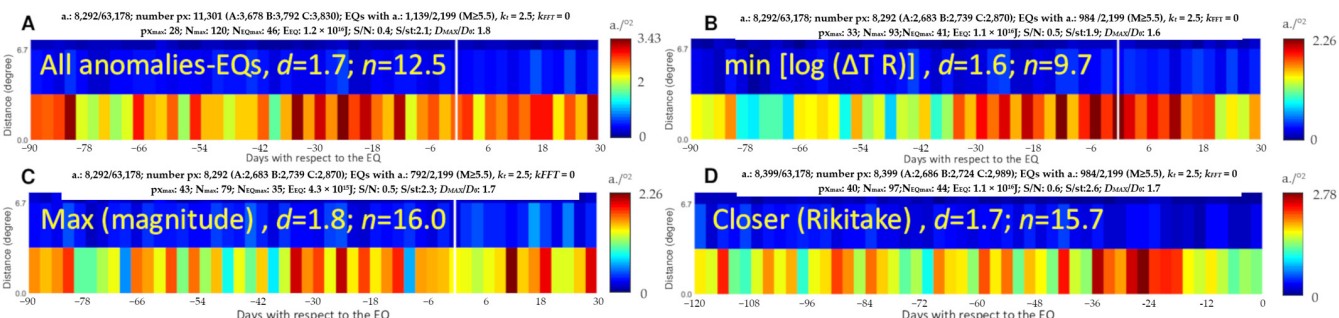

**Figure 1.** Correlation of *Swarm* magnetic field anomalies with M5.5+ earthquakes using four different methods: (**A**) all anomalies are associated with all compatible earthquakes; (**B**) each anomaly is associated with the closest earthquake in space and time, i.e., the minimum of log(ΔT R); (**C**) the anomaly is associated with the greatest magnitude earthquake; (**D**) the anomaly is associated with the earthquake closer to the Rikitake law, with coefficients from De Santis et al. [29]. The graphs are cut for the vertical section without significant information, showing only the closer band to the earthquake epicenter. The color bar represents the density of anomalies in terms of the number of anomalies per square degree.

When the concentration of the real anomalies is compatible with a homogeneous random simulation of anomalies, we expect $d \cong 1$ and $n \cong 0$. At the same time, we define a statistically significant concentration, i.e., a real distribution of anomalies with the largest concentration significantly differing from a random distribution, if $d \geq 1.5$ and $n \geq 4.0$, as suggested by De Santis et al. [29].

## 3. Results

We consider the three satellites of the *Swarm* mission and analyze two parameters: the Y-East component of the magnetic field and the electron density, *Ne*. All these results are shown in the following four subsections. Here, we present the figures with the most interesting results, while we provide other figures in the Supplementary Materials.

### 3.1. Swarm Magnetic Field Results

In this section, we applied the WSC algorithm to the *Swarm* magnetic field data recorded by the three main satellites of the constellation from 26 November 2013 until 25 November 2021, i.e., the first 8 years of available data. Figure 1 represents the WSC applied to the *Swarm* Y-East component magnetic field anomalies with M5.5+ earthquakes in a time window of 120 days (i.e., 90 days before each earthquake and 30 after for methods

1, 2, and 3, and 120 days before each earthquake for method 4) using the four methods to select which earthquake is associated with the anomaly in case of ambiguity. We note that all the methods identify the same group of anomalies from around −40 to −10 days, and, in some of them, it is also the absolute maximum concentration. It is very interesting that, even with Method 2, which minimizes the log (ΔT R), the absolute maximum concentration is 12 days prior to the earthquake and not the co-seismic, as the method would forcefully enhance the anomalies closer to the event (actually, it is only this method that presents a significant concentration at the earthquake occurrence, although it is not the largest one).

Figure S2 (in Supplementary Materials) reports the statistical correlation analysis for the *Swarm* Y-East magnetic field component anomalies (extracted with a threshold $k_t = 2.5$) with M5.5+ earthquakes from 500 days before until 500 days after each event using Method 1. With these criteria, we can obtain symmetric analyzed time and see that the maximum concentration of anomalies preceded the earthquake (of about 30 ± 10 days). Furthermore, the concentration of anomalies before the earthquake is slightly higher than that after the event. The total number of anomalies in the first row (i.e., the closest to the epicenter) before the earthquake is 16,630, while the total number after the earthquake is 16,436, showing that, before the earthquake, there are 194 more anomalies than the ones after it. Even if this number is just 1.2% more, it could support the idea that, inside the identified anomalies, there could be a part likely produced by ionospheric seismo-induced phenomena. A specific discussion that took into account the geomagnetic quiet and disturbed amount of time before and after each event is reported in Section 4.5, and the specific difference seems to be influenced by this aspect.

To search for a possible relationship between the anticipation time and magnitude of the earthquakes, the events have been divided into five magnitude bands (the same used in Figure 5 of De Santis et al. [30]): 5.5 ≤ M ≤ 5.9, 6.0 ≤ M ≤ 6.4, 6.5 ≤ M ≤ 6.9, 7.0 ≤ M ≤ 7.4, and M ≥ 7.5 using Method 1. Figure 2 represents the WSC applied to 8 years of *Swarm* magnetic field anomalies and earthquakes divided into magnitude bands. Here, an extended period before the earthquake (1000 days instead of 500, as in [29]) is used, as now the statistics are better supported by a larger amount of data. It is worth noting that, in these analyses, a part of the events is not fully analyzed. In particular, the earthquakes that occur in the first 1000 days of the analyzed years have an incomplete time window before them.

In Figure 2B, we can see that, in the four larger magnitude bands (i.e., for M6+), the absolute maximum concentration increases perfectly with the magnitude. The anticipation time in the function of the magnitude is presented with a fit (as determined by the Rikitake law, i.e., $\log \Delta T = a + bM$) in Figure 2A, where $a = -0.96 \pm 0.73$, and $b = 0.51 \pm 0.10$. For the lower magnitude earthquake bands (i.e., 5.5 ≤ M ≤ 5.9), the maximum concentrations are at −210, −150, and −90 days, but at the beginning of the graph (where we expect the maximum concentration corresponding to this band), there is also a high concentration of anomalies. We can explain such results in different ways, either independently or occurring simultaneously. One explanation is based on the fact that the lower magnitude earthquakes, i.e., those below M6.0, are less likely to produce distinguishable LAIC effects, while only a part of them can still show some possible ionospheric earthquake precursors. Furthermore, eventual ionospheric electromagnetic perturbations from lower magnitude earthquakes could be very much weaker, and the method presented in this paper could be insufficiently sensitive to detect all of them. An alternative explanation is that these lower magnitude earthquakes are so frequent that an anomaly can be associated with many of them, including those with a longer anticipation time. A third possible reason is that this lower magnitude band could have a typical anticipation time of less than 20 days, which is the most affected by the removal of earthquakes at these distances from larger earthquakes due to the applied declustering method. Despite these considerations, we noted that, in the Rikitake law of Figure 2A, the "not aligned" magnitude band with the fit is the one with earthquakes in the range of 6.0 ≤ M ≤ 6.4. Future investigations are necessary to clarify such ambiguity.

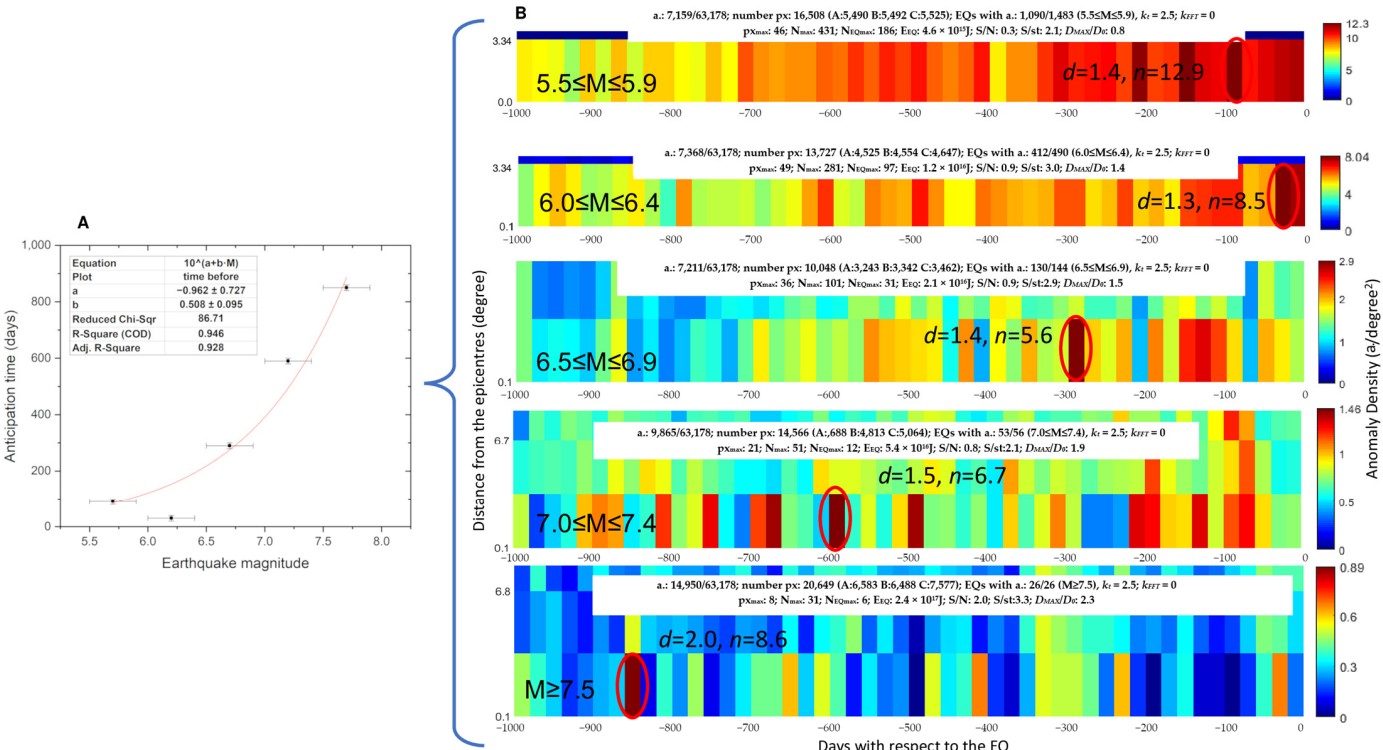

**Figure 2.** Correlation of *Swarm* magnetic field anomalies with earthquakes divided into five magnitude bands. (**A**) Fit of the anticipation time of maximum concentration (underlined by a red circle in subfigure (**B**)) as a function of earthquake magnitude using the equation $\Delta T = 10^{a+b \cdot M}$. The error bar takes into account the width of the pixel. (**B**) Details of the analysis in the earthquake magnitude bands of $5.5 \le M \le 5.9$, $6.0 \le M \le 6.4$, $6.5 \le M \le 6.9$, $7.0 \le M \le 7.4$, and $M \ge 7.5$. Only the significant part of the superposed space-time graph is represented. The statistical indications of $d$ (how many times the maximum real concentration is higher with respect to the random one) and $n$ (how many standard deviations there are in which the real largest concentration is greater than the random one) are also shown. Such maximum concentrations (as objectively identified by the WSC algorithm) have been highlighted by a red or orange circle (the last one for a less significant concentration in terms of the Rikitake law). The color bar represents the density of the anomalies in terms of the number of anomalies per square degree.

A summary view of the WSC analysis of the *Swarm* magnetic field data applied to all the investigated M5.5+ earthquakes is provided in the Supplementary Materials in Figure S3, showing higher concentrations toward the earthquake occurrence.

To validate such results, an earthquake catalog with a shuffled magnitude has been produced, and we showed the results in Figure S4 in the Supplementary Materials. In the subplot on the left, it is possible to see that a correlation between the shuffled earthquake magnitudes and the real set is practically absent (R = 0.0004), while, on the right side of Figure S3, it is shown that the relationship between the anticipation time and the earthquake magnitude is lost, as expected after shuffling the earthquake catalog. Some minor correlations can still be found, but this could simply be due to the fact that, since we shuffled only magnitudes, the epicenter and the origin time of the events are still real, so the anomalies really preceded an earthquake.

Figure 3 shows the different analyses of the *Swarm* Y-East component magnetic field data, taking into account the signal's frequency correlated with M5.5+ earthquakes. For these analyses, we show the classical WSC with Method 1 (all possible combinations of anomalies and earthquakes) from 500 days before until 500 days after the earthquake. The periods are from the upward shorter period (2–10 s) (subpanels A) to the downward longer period (25–50 s) (subpanel C). We can note that it seems that the anticipation time increases

with the period of the signal, this being the time between the maximum concentration of the anomalies and the occurrence of the earthquake. This observation could be crucial for a future system to predict earthquakes, as the frequency of the signal could give important information of the time-lapse to the earthquake. We tried to provide a rough linear fit (see Figure S5 in the Supplementary Materials) of such an anticipation time, obtaining:

$$\Delta T = -24\ (\pm 19) + 4.01\ (\pm 0.82) \cdot P \tag{4}$$

where "$\Delta T$" is the anticipation time in days, and "$P$" is the period of the signal in seconds.

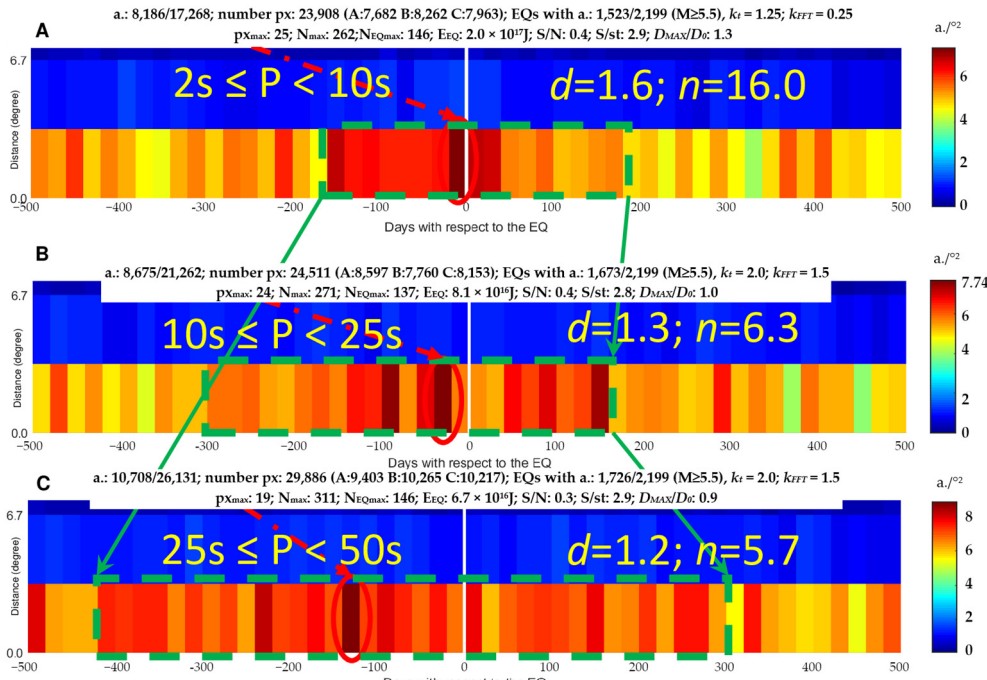

**Figure 3.** 8-year *Swarm* Y-mag data separated for frequency content in the anomalies correlated with M5.5+ earthquakes from 500 days before until 500 after their occurrence. The color bar represents the density of the anomalies in terms of the number of anomalies per square degree. The anomalies have been selected with higher signal content in three different period bands: (**A**) from 2 s to 10 s, (**B**) from 10 s to 25 s and (**C**) from 25 s to 50 s.

The fit provides a positive slope of about 4.0 days/s. The fact that the fit provides a negative intercept and then a possible negative anticipation time (and thus an effect after the earthquake) could be explained by the fact that the period stops at about 6 s before the earthquake's occurrence (i.e., $\Delta T = 0$, then $P = 6$ s). We provide a specific discussion on these results in Section 4.4.

We finally noted that there seems to be a very long period underlined by dashed green boxes in Figure 3, with higher anomaly concentrations with respect to the rest (the identification of the green boxes is taken by sight considering the "more red or brown" bins, subjectively). We speculate that the density of the anomaly in the tails (i.e., before or after the green dashed box) is not due to the earthquake and that it is in the background due to other sources, while the increase in the anomalies could be related to a sort of seismic activation and to the consequences of the seismic event after their occurrence. Even these higher anomaly regions increase their duration (before the earthquake) with the period of the investigated signal. Future studies are fundamental to investigate this aspect better.

Table S1 (in the Supplementary Materials) summarizes the main statistical results for all the investigated combinations of the 8-year *Swarm* magnetic field data correlated with M5.5+ shallow earthquakes. In particular, the two statistical quantities $d$ and $n$ are reported, along with the indication of when the maximum concentration of anomalies

occurs in the bin closest to the epicenter (negative if it occurs before the earthquake and positive if it happens after) and how many earthquakes are associated with anomalies in the pixel with the maximum concentration of anomalies. When a concentration passed the statistical evaluation criteria defined in [29] (i.e., $d \geq 1.5$ and $n \geq 4.0$) and occurred before the earthquake, we underline the row in green.

### 3.2. Swarm Electron Density Results

In this section, we applied the WSC algorithm to the electron density (*Ne*) data recorded by the three main satellites of the *Swarm* constellation from 2 December 2013 until 1 December 2021, i.e., the first 8 years of available data. Most of the results of *Ne* are provided in the Supplementary Materials, as they have not been found to be statistically significant. Figure S6 (in the Supplementary Materials) provides the results of WSC applied to 8 years of *Swarm Ne* data correlated with M5.5+ shallow earthquakes using the above four methods to associate the anomalies and earthquakes. The results do not pass the statistical criteria defined in [29]: $d \geq 1.5$ and $n \geq 4.0$. Furthermore, the analyses with the minimization of log ($\Delta$T R) and the maximum earthquake magnitude (methods 2 and 3) find a post-seismic bin as the maximum concentration. In fact, the analysis in the long symmetrical time from 500 days before until 500 days after the earthquakes with the same data (Figure S7 in the Supplementary Materials) shows low or null significance, and, overall, the number of anomalies after the earthquake (15,618) is greater than those before the earthquake (15,504); their difference being equal to 114 anomalies, which is lower than the square root of the anomalies, which is equal to 125 and could be considered as an approximation of their uncertainty, as further discussed in Section 4.4.

In the same way that we detected an increase in the anticipation time of the anomalies with the earthquake magnitude in the magnetic field data (Figure 2), we now apply the same approach to 8 years of the *Swarm Ne* data, as shown in Figure S8 in the Supplementary Materials. In this case, we tried to make a fit through the maximum concentrations of anomalies detected by the WSC algorithm in the first row (i.e., closer to the earthquake epicenter) of different magnitude ranges following the empirical Rikitake law. We note that the magnitude 7.5+ earthquakes analysis shows the absolute maximum concentration on the second row, i.e., between 380 km and 750 km from the epicenter. As such, a magnitude range is the one with fewer earthquakes (26 in total); this effect could be due to an insufficient number of cases to construct good statistics. Furthermore, it is important to note that these analyses did not pass the statistical evaluation criteria, and some of them even show a concentration of the same intensity as the one expected for a random simulation ($d \cong 1$) and a negative *n*, indicating a real largest concentration lower than the random one. Even if the concentration of anomalies is not significant, the maximum concentration in the row closer to the epicenter for magnitudes $\geq 6$ seems to follow the Rikitake law, as can be seen in the fit shown in Figure S8A, which provides the following coefficients: $a = -3.5 \pm 2.1$ and $b = 0.83 \pm 0.27$. The maximum concentration in the magnitude band $5.5 \leq M \leq 5.9$ is found 250 days before the earthquake, but the concentration 10 days before the earthquake is also high but lower than the maximum and, in the fit, has been used as the objective maximum extracted by the code. So, we can suppose that there may be some anomalies related to the earthquake, but the "anomaly definition" employed is too weak to extract all possible seismo-induced phenomena, consequently including many anomalies not related to earthquakes.

To improve the anomaly definition, we introduce the analysis of frequency content in the investigated window with the same approach as that one used for the magnetic field data. As can be seen in Figure 4, the significance of the results is improved for most of the analyses, passing the statistical criteria ($d \geq 1.5$ and $n \geq 4.0$). We note that, for the shorter period analysis in the symmetrical period from $-500$ days to 500 days with respect to the earthquake, the maximum concentration of the anomalies is post-seismic, and for the analysis in the signal period range between 10 s and 25 s from $-90$ days to 30 days, the maximum concentration is a "co-seismic" effect. Even if statistically significant, these

results do not contribute to the main purpose of this paper, i.e., a better understanding of the preparatory phase of medium-large earthquakes. On the other hand, the investigation in the lower frequency band in which the longest period is equal to 25–50 s shows not only highly significant results but also pre-seismic maximum concentrations of anomalies both for the investigation in the time window from −90 days to 30 days and for the one from −500 days to 500 days at about −59 days and −290 days, respectively (see Table S2). Such a result seems to indicate that taking the frequency content of the anomalies into account could help to discriminate the ones possibly caused by the preparation phase of earthquakes from the other ones caused by other phenomena.

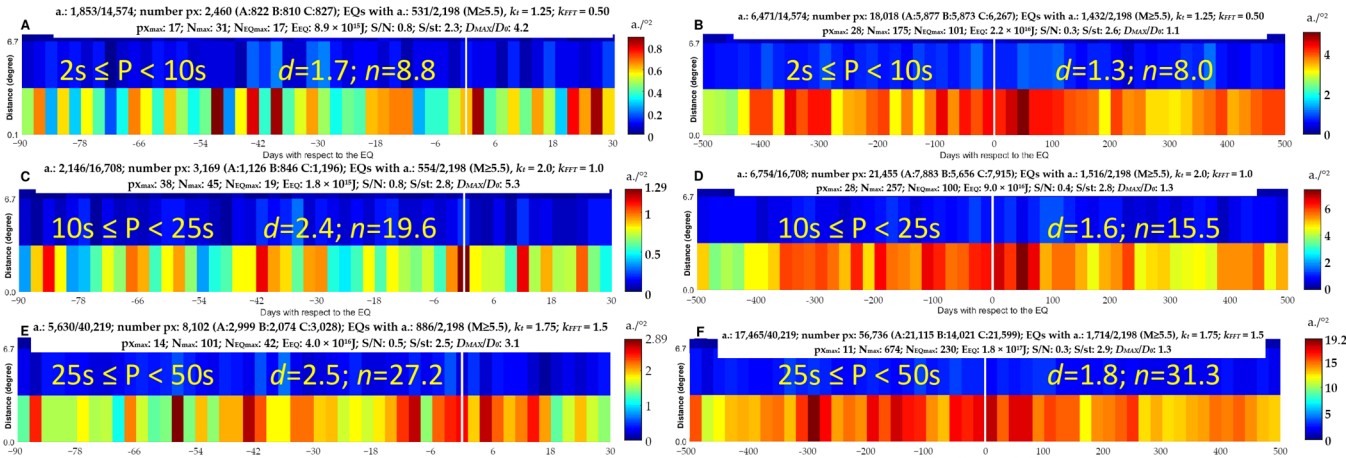

**Figure 4.** 8-year *Swarm* Ne data WSC analyses separated for frequency content in the anomalies correlated with M5.5+ earthquakes for 120 days (left) and 1000 days (right) around the EQ occurrence, in particular: (**A**) signal period from 2 s to 10 s and analysis from −90 days to +30 days with respect to the earthquake, (**B**) signal period from 2 s to 10 s and analysis from −500 days to +500 days with respect to the earthquake, (**C**) signal period from 10 s to 25 s and analysis from −90 days to +30 days with respect to the earthquake, (**D**) signal period from 10 s to 25 s and analysis from −500 days to +500 days with respect to the earthquake, (**E**) signal period from 25 s to 50 s and analysis from −90 days to +30 days with respect to the earthquake, and (**F**) signal period from 25 s to 50 s and analysis from −500 days to +500 days with respect to the earthquake. The color bar represents the density of the anomalies in terms of the number of anomalies per square degree.

Finally, Table S2 (in the Supplementary Materials) reports the main statistical parameters for all the performed analyses for 8 years of *Swarm Ne* data. The statistically significant results (i.e., the ones with $d \geq 1.5$ and $n \geq 4.0$) that precede the earthquake are underlined in green, as in Table S1. It is possible to note that just three out of sixteen results for *Ne* passed these criteria. Despite this, introducing the analysis of the signal frequency provides more significant results in the lowest frequency band (i.e., the period between 25 s and 50 s).

## 4. Discussion

### 4.1. General Comparison for Magnetic Field and Ne Results

Comparing the results of the electron density and magnetic field measured aboard the satellite, it is possible to note that the *Swarm Ne* results are not statistically significant. One reason for this could be the large corresponding number of detected anomalies, which is about 80,000. It will be interesting in future works to apply the same technique to CSES data, as, due to the satellite's Sun-synchronous orbit, it would permit to have *Ne* deep local nighttime (02:00 LT) data, always avoiding the effects of the solar irradiation (for example after sunset, when the ionosphere can show perturbation even until midnight or little later [38]). Therefore, the fixed local time reduces the signal variation in *Swarm* due to its gradual orbital precession. In addition, the crossing of critical local times close to

the terminator can create perturbations or even false anomalies due to the shadow/light passage of the *Swarm* satellite along the same orbit.

*Swarm* satellites show better results for the magnetic field. This could be due to the intrinsically higher quality of magnetic instrumentation in comparison with that of Electric Field Instruments and to the satellite design (in particular, the star cameras on the same optical bench as the fluxgate magnetometers). Physically, this could also be explained as the magnetic field is usually less affected by the local time than *Ne*. Hence, its precession in *Swarm* satellites (see the discussion above) seems not to affect the usability of its measurements to search for possible pre-earthquake ionospheric disturbances.

In addition, a specific comparison of the coefficients found by the empirical Rikitake law in this work for the *Swarm* magnetic field and electron density data and the ones found by De Santis et al. [29] is provided in Figure S9 of the Supplementary Materials. We confirm the previous results and find that almost all the results are within the trend proposed in 1987 by Rikitake [37] for earthquake precursors of the quasi first kind (i.e., earth-currents, resistivity, radon, underground water, and other geochemical elements) or only the geomagnetic precursor (from the ground). This result is promising because it means that, with the availability of new satellite data, we are confirming an empirical trend obtained decades ago for ground precursors and extending its validity to ionospheric precursors, as recently found in De Santis et al. [29].

### 4.2. Validation of the Results by Confusion Matrix Performance Evaluation and ROC Curves

The results have been evaluated by the so-called "confusion matrix", which is a table of two rows × two columns used to assess the performances of a prediction model, crossing the presence (or not) of an ionospheric anomaly followed (or not) by an earthquake in the given time within the pre-chosen distance and time.

Here, we considered the same anomalies presented in the results, but we evaluated a cell that has 6° geomagnetic latitude for 6° geographic longitude and 2.4 days (for investigations previously conducted 90 days before the earthquake) or 20 days (for investigations conducted 500 days before the earthquake). The evaluation of the cell has been selected among the classic four cases:

1. True positive (*TP*): in the cell, there is at least an anomaly, and an earthquake follows within the next 90 days
2. True negative (*TN*): in the cell, there are no anomalies, and no earthquakes follow in the next 90 days in the same cell.
3. False positive (*FP*): in the cell, there is one or more anomalies, and no earthquakes follow in the next 90 days in the same cell
4. False negative (*FN*): in the cell, there are no anomalies, and an earthquake follows in the next 90 days.

From the evaluation of the number of true and false positives and negatives, the accuracy "*Acc*" has been computed as the total number of good cases over the total cases:

$$Acc = \frac{TP + TN}{TP + TN + FP + FN} \tag{5}$$

Better performance with respect to a random prediction is achieved when *Acc* > 50%. We also calculated the Hit Rate (*HR*) and False Positive Rate (*FR*); the former expresses the percentage of predicted events, while the latter is the rate of false alarms:

$$HR = \frac{TP}{TP + FN} \quad FR = \frac{FP}{FP + TN} \tag{6}$$

An ideal prediction system would give the limit case with *HR* = 100% (all events are predicted) and *FR* = 0% (no false alarms), while a real system is better as close as possible to such limit case, and, typically, the earthquake prediction systems are not suitable for

use with the population when the false positive rate is too high. Finally, the Alarmed Time-Space "*AT*" (also called the alarm rate in the literature [39]) has been estimated:

$$AT = \frac{TP + FP}{TP + TN + FP + FN} \tag{7}$$

This quantity is important to check that a good prediction is not just achieved regarding an analyzed space-time that is too large or even an entire space-time. For example, the affirmation that "*in the future, in the World, an M5.5+ earthquake will occur*" is surely true, but it is not a "prediction", because it does not predict the exact time, and the space domain is too large (the whole world). Given the same accuracy, a prediction is preferable with a lower *AT*. The alarmed time can also be compared with the number of positive cases (*TP* + *FN*) with respect to the total cases, as, even an ideal (so, perfect) prediction needs to alarm at least all of the space-time that any event occurred.

Table 1 reports the evaluation of performances in terms of confusion matrices for the *Swarm* magnetic field and electron density data considering 90 days and 500 days before the M5.5+ earthquakes. These tables reflect the analyses shown in Figures 1, 3, 4, S2, S4 and S5, even if we consider only the time before the earthquakes, as the confusion matrix is a tool to assess the prediction capability of a system, so only the ionospheric anomalies that preceded the earthquake can be "positively" evaluated. It is possible to note that all the analyses show a high accuracy, from 72% until 95%. Contrariwise, the hit rates tend to be rather low (i.e., from ~1% to ~19%), as several earthquakes are not preceded by anomalies in the considered interval of time, and, overall, several earthquakes did not show anomalies continuously in the 90 (or 500) days before their occurrence. The alarmed space-time windows are very low (i.e., around one or a few percent) for all the investigations with an alarm-window time of 90 days, and they are between 5.1% and 19.1% for the analyses performed in a time window of 500 days. Both the investigated time windows show alarmed times below the 50% threshold—in some cases, lower than 1%—which provide further evidence of the non-chance nature of the detected anomalies in relation to the following earthquakes. The accuracy is similar to that of a previous work performed on *Swarm* electromagnetic data and earthquakes but using a machine learning approach [40]. Furthermore, the overall performances reflected the previous considerations, with the scores for the magnetic field being generally better than those for the electron density. Further, better scores were always yielded when the frequency content of the signal was investigated, confirming that it is fundamental to take into account the signal frequency of the possible electromagnetic ionospheric precursor in future studies.

In order to further statically check the predicting capability of the magnetic field and electron density anomalies extracted in this work as the best candidates for earthquake precursors by WSC, the Receiving Operating Characteristic (ROC) graph has been calculated. ROC is a well-established method to test a precursor candidate, e.g., [41], and it is a graph where the false positive rate is conventionally reported on the horizontal axis and the hit rate is reported on the vertical one. The diagonal from (0%, 0%) to (100%, 100%) represents the random case, while the upper-left corner (0%, 100%) is an ideal case that predicts all the events without any false alarm. A good prediction is required to be as far from the diagonal toward the ideal case as possible.

In our case, we extract all the anomalies identified as statistically significant in this work, i.e., the pre-earthquake concentrations that show a statistical coefficient $d \geq 1.5$ and $n \geq 4.0$, as previously used in this work and first defined by De Santis et al. [29]. In this section, other concentrations that are different from the absolute maximum one that passed this statistical criterion have been also taken into account. Collecting all the anomalies from the magnetic field and electron density *Swarm* data and all the techniques, we produced a new dataset of the best anomalies that contains 4831 *Swarm* anomalous signals after cleaning the overcounted ones.

With the new dataset of the best WSC anomalies, the "Confusion matrix" and ROC diagram have been evaluated and reported in Table 2 and Figure 5, respectively. The step

time of calculus has been set to be equal to the alerted time, and the window following the one with any *Swarm* electromagnetic anomaly is alerted if it contains any anomaly; otherwise, it is not alerted. Comparing Table 2 with Table 1, it is possible to fully confirm that the selections made in this paper extracted the statistically best pre-earthquake candidates permitted to reduce the false positive rate and increased the hit rate.

**Table 1.** Confusion matrices of the *Swarm* magnetic field and electron density anomalies identified in the 90 days or 500 days before the M5.5+ earthquakes.

**90 days before the earthquake**

| | | Magnetic Data | | Magnetic Data 2–10 s | | Magnetic Data, 10–25 s | | Magnetic Data, 25–50 s | |
|---|---|---|---|---|---|---|---|---|---|
| | | Earthquake | | Earthquake | | Earthquake | | Earthquake | |
| | | Yes | No | Yes | No | Yes | No | Yes | No |
| Anomalies | Yes | 1396 | 25,654 | 573 | 9477 | 555 | 10,453 | 655 | 13,149 |
| | No | 50,454 | 1,091,776 | 51,277 | 1,107,953 | 51,295 | 1,106,977 | 51,195 | 1,104,281 |
| Hit and false positive rates | | HR = 2.7% | FR = 0.23% | HR=1.11% | FR=0.85% | HR = 1.07% | FR = 0.94% | HR = 1.26% | FR = 1.18% |
| Accuracy and Alarmed Time | | Acc = 93.5% | AT = 2.31% | Acc = 94.8% | AT = 0.86% | Acc = 94.7% | AT = 0.94% | Acc = 94.5% | AT = 1.18% |

**500 days before the earthquake**

| | | Earthquake | | Earthquake | | Earthquake | | Earthquake | |
|---|---|---|---|---|---|---|---|---|---|
| | | Yes | No | Yes | No | Yes | No | Yes | No |
| Anomalies | Yes | 3255 | 19,575 | 1378 | 7826 | 1522 | 8800 | 1805 | 10,882 |
| | No | 15,325 | 102,965 | 17,202 | 114,714 | 17,058 | 113,740 | 16,775 | 111,658 |
| Hit and false positive rates | | HR = 17.5% | FR = 16.0 | HR = 7.42% | FR = 6.39% | HR = 8.19% | FR = 7.18% | HR = 9.71% | FR = 8.88% |
| Accuracy and Alarmed Time | | Acc = 75.3% | AT = 16.2% | Acc = 82.2% | AT = 6.52% | Acc = 81.7% | AT = 7.31% | Acc = 80.40% | AT = 8.99% |

**90 days before the earthquake**

| | | *Ne* data | | *Ne* data 2–10 s | | *Ne* data, 10–25 s | | *Ne* data, 25–50 s | |
|---|---|---|---|---|---|---|---|---|---|
| | | Earthquake | | Earthquake | | Earthquake | | Earthquake | |
| | | Yes | No | Yes | No | Yes | No | Yes | No |
| Anomalies | Yes | 1380 | 32,605 | 345 | 7984 | 434 | 7460 | 1100 | 16,156 |
| | No | 50,502 | 1,084,793 | 51,537 | 1,109,414 | 51,448 | 1,109,938 | 50,782 | 1,101,242 |
| Hit and false positive rates | | HR = 2.66% | FR = 2.92% | HR = 0.66% | FR = 0.71% | HR = 0.84% | FR = 0.67% | HR = 2.12% | FR = 1.44% |
| Accuracy and Alarmed Time | | Acc = 92.9% | AT = 2.91% | Acc = 94.9% | AT = 0.71% | Acc = 95.0% | AT = 0.68% | Acc = 94.3% | AT = 1.48% |

**500 days before the earthquake**

| | | Earthquake | | Earthquake | | Earthquake | | Earthquake | |
|---|---|---|---|---|---|---|---|---|---|
| | | Yes | No | Yes | No | Yes | No | Yes | No |
| Anomalies | Yes | 3436 | 23,490 | 949 | 6744 | 1039 | 6101 | 2311 | 11,662 |
| | No | 15,164 | 99,030 | 17,651 | 115,776 | 17,561 | 116,419 | 16,289 | 110,858 |
| Hit and false positive rates | | HR = 18.5% | FR = 19.2% | HR = 5.10% | FR = 5.50% | HR = 5.59% | FR = 4.98% | HR = 12.4% | FR = 9.52% |
| Accuracy and Alarmed Time | | Acc = 72.6% | AT = 19.1% | Acc = 82.7% | AT = 5.45% | Acc = 83.2% | AT = 5.06% | Acc = 80.2% | AT = 9.90% |

**Table 2.** Confusion matrices of the best *Swarm* magnetic field and electron density anomalies identified in the WSC concentrations that passed the statistical significance test (i.e., the ones with $d \geq 1.5$ and $n \geq 4.0$).

| | | Alerted Time of 90 Days | | Alerted Time of 500 Days | |
|---|---|---|---|---|---|
| | | Earthquake | | Earthquake | |
| | | Yes | No | Yes | No |
| Anomalies | Yes | 564 | 1112 | 406 | 478 |
| | No | 800 | 29204 | 258 | 4618 |
| Hit and false positive rates | | HR = 41.35% | FR = 3.67% | HR = 61.15% | FR = 9.38% |
| Accuracy and Alarmed Time | | Acc = 94.0% | AT = 5.29% | Acc = 87.2% | AT = 15.35% |

The ROC curve in bold blue in Figure 5 is obtained after testing several alerted times from 10 days (lower left side) to 2500 days (upper side). Two black arrows have underlined the two alerted times of 90 and 500 days, and they correspond to the one used in Table 2 and are mostly selected in this paper. In the same figures, the Area Under the Curve (AUC) has also been represented as a black inclined hatch for the random case (i.e., the one under the diagonal equal to 50% for the definition) and as a red inclined hatch for the real best *Swarm* anomalies, estimated to be equal to 83.6%. So, the AUC of the real data is 1.67 times better than that of a random predictor, confirming that WSC applied to *Swarm* data is able to extract potential pre-earthquake ionospheric disturbances.

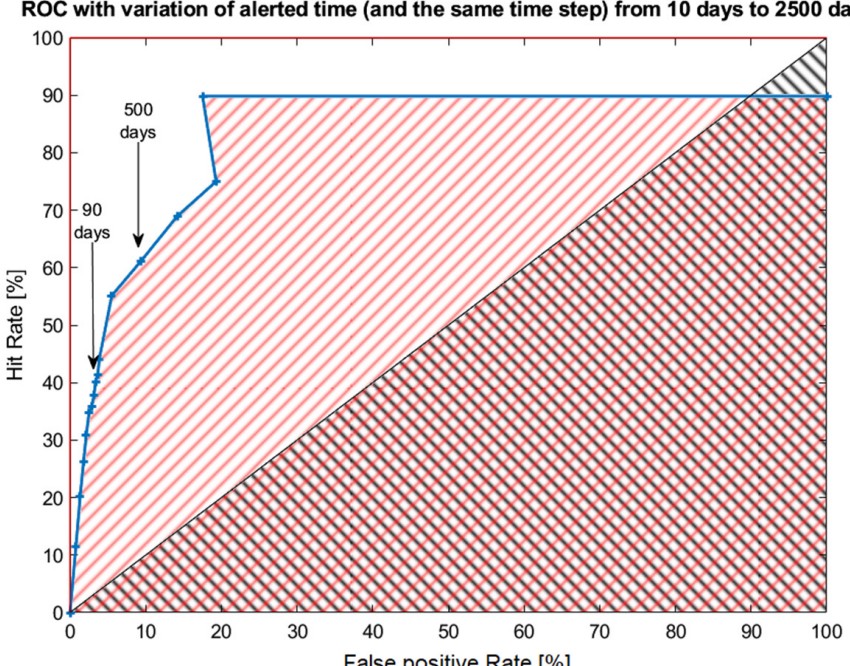

**Figure 5.** Receiving Operating Characteristic (ROC) graph for the best *Swarm* magnetic field and electron density anomalies (i.e., the ones in WSC concentrations with significant statistical parameters $d \geq 1.5$ and $n \geq 4.0$) calculated, varying the alerted time from 10 days to 2500 days. The Area Under the Curve (AUC) is represented as a black dashed pattern for a random case and as a red dashed pattern for the real *Swarm* case.

### 4.3. Improving the Superposed Epoch Approach by Taking into Account the Rikitake Law

The results of this paper, especially for the magnetic field anomalies, strongly confirm the empirical Rikitake law [37], which predicts that the logarithm of anticipation time linearly increases with earthquake magnitude. If such a law is true, then the superposed epoch approach would not work due to the mixing of several earthquakes with different magnitudes. It seems to work in the real case, but we think that this is due to the natural distribution of the earthquake magnitudes that follows the Gutenberg–Richter [42] earthquake distribution, privileging the smaller magnitude events. In this paper, we propose a new approach to addressing this issue when the epochs are superposed by "normalizing" the time scale using the Rikitike law, with the coefficients (*a* and *b*) being those proposed by De Santis et al. [29] for the *Swarm* magnetic field and electron density satellite data and referring to an M6.5 earthquake, following the next equation:

$$\Delta T' = \Delta T \cdot \frac{10^{(a+b\cdot6.5)}}{10^{a+b\cdot M}} \tag{8}$$

where $\Delta T'$ is the normalized anticipation time, $\Delta T$ is the original time with respect to the earthquake, and *M* is the magnitude of the earthquake under analysis. Even if the formula can be, in principle, also applied to post-earthquake anomalies, we think that it is not proper to use it in such a way, as the formula is constructed on possible precursors. The reference magnitude (here, chosen as M6.5) can be, in principle, any magnitude, and the results are not affected by this choice. Its scope is to report the time in a different scale chosen as the number of days that preceded an M6.5 earthquake. M6.5 is an intermediate magnitude with a high number of events in the investigated period to be able to clearly show possible precursors, according to the results illustrated in this paper.

To select the window of analysis, we look at the results from [29] and plotted the expected anticipation time as a function of the earthquake magnitude (in Figure 6, as the orange curve). Based on these data, we decided to analyze 150 days before the earthquake

for the magnetic field data and 85 days for the *Ne* measurements (when *M* = 6.5, i.e., our reference point). We would underline that, using Equation (4), the analyzed time before the earthquake is not fixed (as in the previous analyses), but it is different for any earthquake as a function of its magnitude, and it is represented as the blue curve in Figure 6, i.e., the result of applying Equation (4)—that is, how many days the normalized time corresponds for other magnitudes. For example, the analyzed time before an M8.0 earthquake is 4037 days for the magnetic field (or 1576 days for *Ne*) according to Equation (4), and the time interval from 4037 days before and until the event will be compressed in the horizontal axis (of Figures 7 and 8) and shown to be equivalent to 150 days to 0 days of an M6.5 earthquake. We note that, for the magnetic field, the larger magnitude earthquakes (M > 8) could show an anticipation time that is greater than that of the 8 years of *Swarm* data available at the moment (the red dashed horizontal line in Figure 6).

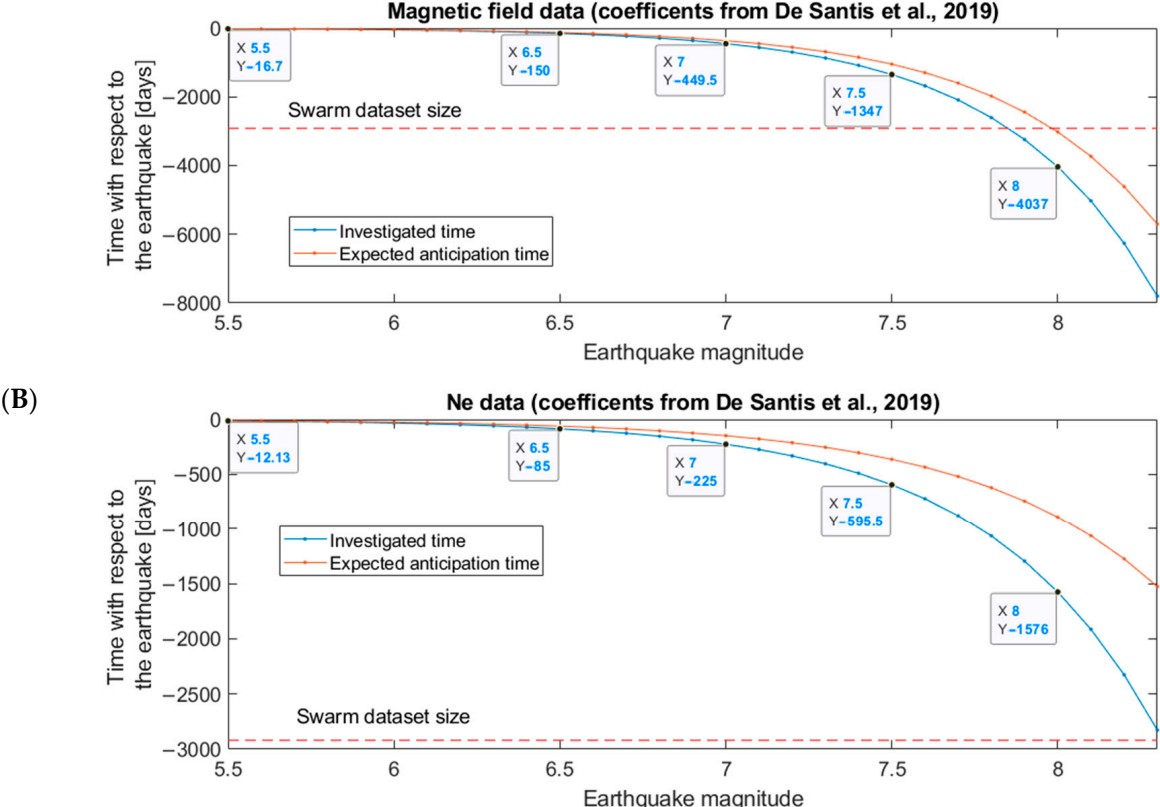

**Figure 6.** Expected anticipation time of magnetic field (**A**) and *Ne* (**B**) anomalies with respect to the earthquake, using the Rikitake law [37] and considering the coefficients found by De Santis et al. [29] for 4.7 years of *Swarm* data (orange line). The investigated time window is shown as a blue line, and the 8-year *Swarm* dataset maximum size is shown as a dashed red line. The data tips provide the time that has been analyzed before a specific magnitude earthquake, applying Equation (4).

We test and apply this approach to the most significant results of the *Swarm* data, which are the investigations of the magnetic field and *Ne* data with Method 1 and the *Swarm* magnetic field anomalies in the band of 2–10 s and the *Ne* anomalies in the band of 25–50 s, which are those with the best statistical indications from the previous analyses.

Figure 7 shows the results for the *Swarm* data without taking into account the frequency of the signal. The maximum concentration has a higher statistical significance with respect to the previous analyses: in fact, for the magnetic field data, *d* passes from 1.7 to 2.1, and for the *Ne* data, *d* passes from 1.3 to 1.5, which also means that, from a non-statistically

significant result, we obtain a higher significance with this different superposed time approach, which implies that we could better investigate the identified anomalies to search for seismo-ionospheric disturbances.

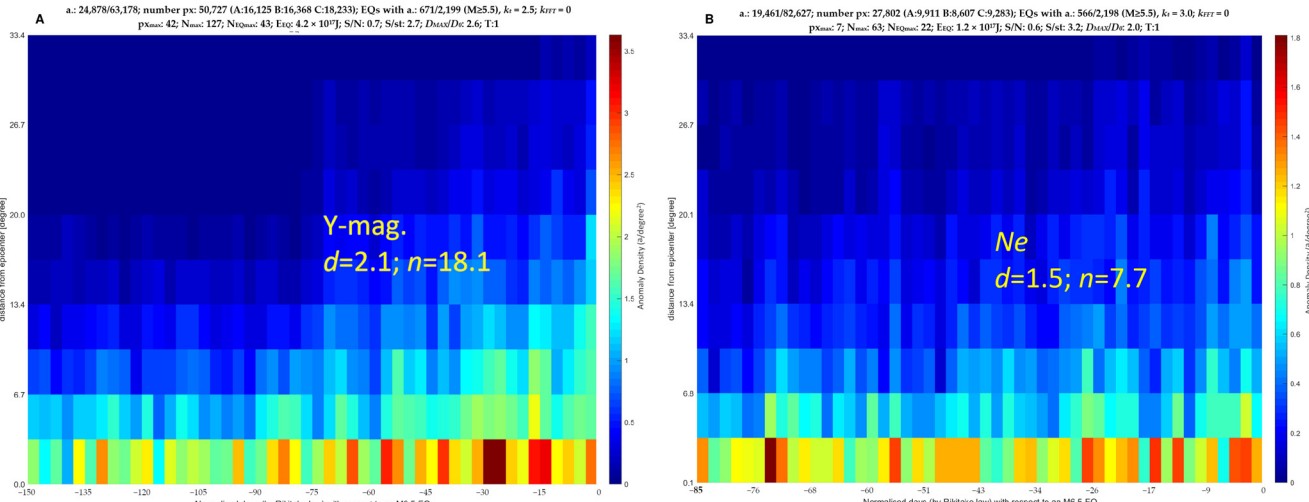

**Figure 7.** *Swarm* magnetic field data (Y-mag in (**A**)) and electron density (*Ne* in (**B**)) correlated with M5.5+ earthquakes using modified time according to the empirical Rikitake law.

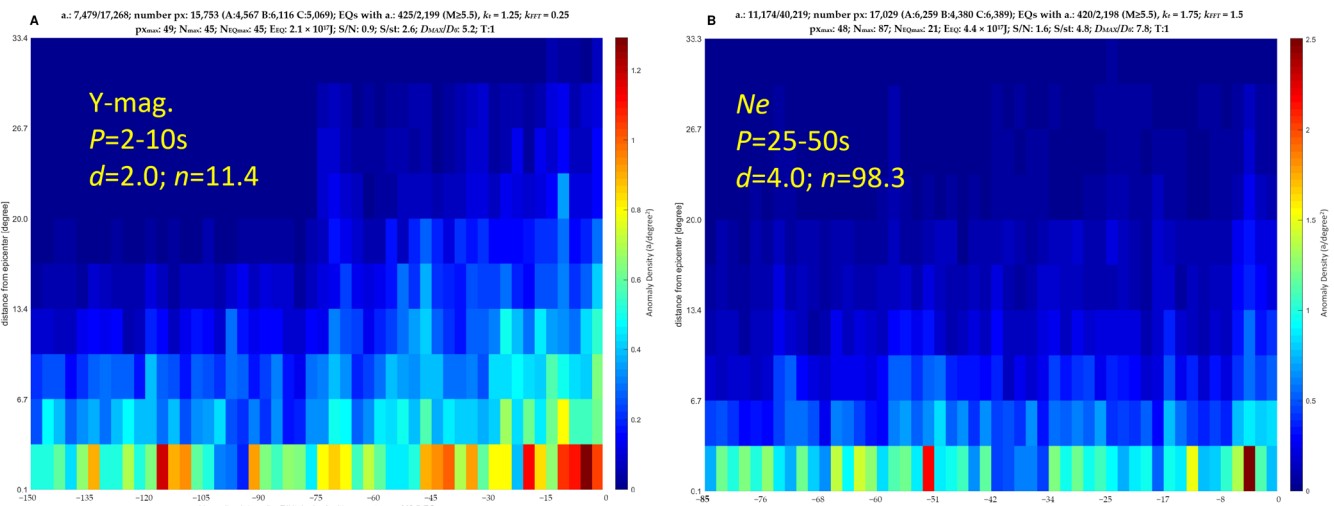

**Figure 8.** *Swarm* magnetic field data in the shorter period of 2–10 s (Y-mag in (**A**)) and electron density in the longer period of 25–50 s (*Ne* in (**B**)) correlated with M5.5+ earthquakes using modified time according to the empirical Rikitake law.

In the same way, in Figure 8, we apply this approach to the anomalies extracted by analyzing the frequency content of the magnetic field and *Ne* data. The frequency selected is the best from the previous analysis, so we obtained the magnetic field data in the quicker signal (2–10 s) and the *Ne* data in the slower signals (25–50 s). In this case, there is a general gain in the statistical improvements of both the *d* and *n* parameters (for the magnetic field data: *d* passes from 1.8 to 2.0, and *n* passes from 9.5 to 11.4; for the *Ne* data, *d* passes from 2.5 to 4.0, and *n* passes from 27.2 to 98.3, recording a very strong increase in statistical significance).

Finally, we underline that this approach seems very promising but requires a very large dataset. In fact, for the larger magnitude earthquakes, the whole time is not covered, even with 8 years of satellite data. On the other hand, for the visual inspection of the results, the immediate reading is lost with this approach: for example, the magnetic field data of

Figure 7A show several concentrations of anomalies between 75 days and 1 month before the M6.5 earthquakes, but such time corresponds to a different interval for any earthquake magnitude that can be estimated by Equation (4) and is represented in Figure 6A. However, as the empirical Rikitake law seems to describe the possible precursor well, it is fundamental to consider it and include it in a future prediction platform. In practice, detecting an anomaly can indicate a lower magnitude earthquake or a larger magnitude earthquake at a later time, and, if a lower magnitude earthquake does not happen immediately, after a number of days/weeks, the prediction can be adjusted, increasing the magnitude of the expected earthquake. Unfortunately, the larger number of false alarms (the anomalies not associated with any seismic events) makes it still impossible to implement such a prediction system.

### 4.4. Some Features of the Largest Concentrations

In the results shown in Figure 3, it seems to be observed that the content of the high frequency of the magnetic field signal of the possible earthquake precursor becomes more intense as the earthquake is approaching. There is not a simple explanation for this, and the present theories would suggest the contrary direction. From a seismological point of view, the crack dimension is supposed to increase toward the earthquake (e.g., [43,44]), the duration of the electromagnetic pulses eventually generated by the microfracture is supposed to be proportional to the crack dimension, and the emitted frequency is supposed to be proportional to the inverse of the pulse duration [6,7]. Even supposing that the fault is simplified as a two planar faces capacitor forming an oscillating circuit with the resistance and inductance of the ground (it can be the fluid patches in the rocks that act as a "transmission line" for the higher conductivity of fluids), the frequency "$\omega$" is supposed to decrease with the crack size "$l$", which increases (a similar calculus that found that $\omega \propto \frac{1}{l}$ is provided by Kachakhidze et al. [45] and Liperovsky et al. [21]). These calculations and theories would not support the result presented in this paper, as they point to the opposite conclusion. However, first of all, we think that the problem is really open. Furthermore, we are not sure that the electromagnetic disturbances observed in the ionosphere are directly produced on the fault, as other phenomena can act at the ground-lower atmosphere interface, as suggested by Kuo et al. [10,11], who simulated the formation of ionospheric disturbances with accumulation on the Earth's surface of positive-holes released from the crack in the fault or a very different and complex chain of processes that influence the ionosphere as a consequence of the global atmospheric electric circuit alteration induced by air ionization from radon release, as theorized by Pulinets and Ouzounov [16]. In the last two mechanisms, it would be very difficult for the frequency of the supposed generated signal to be related to any crack evolution model, so it would be complicated to predict if the frequency of the magnetic field signal can increase or decrease toward the earthquake occurrence. Further studies will be necessary both in theory and in the observations, as shown here.

We finally investigated if the earthquakes involved in the absolute maximum concentrations of Figure 3A–C share some common features, such as focal mechanism or the location of the hypocenter beneath the sea or the land. A similar investigation on the *Swarm Ne* data and on earthquake occurrences over sea or land has been reported by He et al. [46]. Even if the authors did not find a particular "preference" in such classification, they found a better correlation for higher magnitude events, as was also detected in the present work.

First of all, we found that the earthquakes that are associated with the three maximum concentrations in the different analyzed signal bands (146 EQs, 137 EQs, and 146 EQs, respectively) are actually different events (except for two that are shared in all three concentrations), so the increase in frequency toward the earthquake occurrence could be apparent, and it is possible that the frequency of the generated signal depends on other parameters such as those investigated in the following. Table 3 reports the number of earthquakes and their percentage in the specific class. The focal mechanism has been

checked, searching for the same earthquake in the global Centroid Moment Tensor (CMT) catalog [47,48] with a maximum tolerance of 2 min on the origin time, 2.5 on the magnitude estimation, and 2 degrees in the epicenter localization. Using one of the rake angles of the moment tensors solution, the focal mechanism has been classified as "reverse" (or "thrust") if $+20° <$ rake $< +160°$, "normal" if $-160° <$ rake $< -20°$, or "strike-slip" if $-20° \leq$ rake $\leq +20°$ or if $|\text{rake}| \geq 160°$, following Cronin [49].

**Table 3.** Investigation of a possible relation with the focal mechanism with the recorded frequency of the magnetic field anomalous signals from the *Swarm* and sea or land epicenter location. The significant percentage of deviation from the standard earthquake distribution (i.e., the ones $\geq 5\%$ in their absolute value) are marked by bold numbers. Green (red) numbers represent positive (negative) deviation percentages with respect to the reference.

| Focal Mechanism | Earthquakes with Anomalies in the Band 2–10 s | Earthquakes with Anomalies in the Band 10–25 s | Earthquakes with Anomalies in the Band 25–50 s | All the Earthquakes |
|---|---|---|---|---|
| strike-slip | **50 (35.2%)** −**10.1%** | 54 (39.4%) +0.6% | **54 (37.2%)** −**5.0%** | 846 (39.2%) |
| reverse | **67 (47.2%)** +**10.5%** | **62 (45.3%)** +**6.0%** | 64 (44.1%) +3.4% | 922 (42.7%) |
| normal | 25 (17.6%) −2.8% | **21 (15.3%)** −**15.4%** | 27 (18.6%) +2.8% | 391 (18.1%) |
| **Sea or land** | | | | |
| Land | **16 (11.0%)** −**35.2%** | 24 (17.5%) +3.6% | **30 (20.6%)** +**21.5%** | 372 (16.9%) |
| Sea | **130 (89.0%)** +**7.2%** | 113 (82.5%) −0.7% | 116 (79.5%) −4.4% | 1827 (83.1%) |

The classification for the focal mechanism and the sea/land localization of the epicenters has been reported in the last column of Table 3 and for all the earthquakes investigated in this study. Further, it needs to be taken as a reference to search for any deviations from these percentages. We report in green (positive) or red (negative) the deviation percentage with respect to the reference, and in bold, we underline the significant difference for the focal mechanisms. Faster signals (2–10 s) seem more likely (+10.5% with respect to their natural distribution) to be produced by reverse focal mechanism earthquakes at the expense of strike-slip ones (−10.1%). The signals with a period in the range 10–25 s seem less likely (−15.4% with respect to their standard distribution) to be produced by normal fault earthquakes, and for the slower signals, the deviation percentages are not greater than 5%, so they are not really significant. Unfortunately, the detected deviations are not statistically significant, and deeper research in the future will be necessary to understand whether the focal mechanism has no influence on the possible ionospheric electromagnetic earthquake disturbances or if our study is too limited to have a clear detection of such dependence. On the other side, land earthquakes are more likely to show lower frequency signals (from faster to slower frequencies, they are −32.2%, +3.6%, and +21.5%), and, complementarily, the sea earthquakes seem likely to produce faster signals (from faster to slower frequencies, they are +7.2%, −0.7%, and −4.4%), as the percentage shows a clear trend with significant deviations for the highest and lowest frequency signals.

Finally, we find that only two earthquakes (above 146 with anomalies in 2–10 s, 137 in 10–25 s, and 146 in 25–50 s) are common events to all of these concentrations, so the results of this analysis are related to different earthquakes, and it is not clear if the frequency of the eventual electromagnetic wave produced by a single earthquake changes with the time, while the focal mechanism seems to have a slight or null influence on the signal frequency, and the localization in sea or land seems to play an important role for the frequency of the anomalies.

### 4.5. General Comparison of the Number of "Pre-" and "Post-" Earthquake Anomalies

Finally, we provide an objective comparison of the number of anomalies detected before and after the earthquake with a symmetrical analyzed time in Table 4. In this

table, we provide the difference between the anomalies detected before and the ones detected after the earthquake. The percentage of such difference refers to the anomalies after the earthquake, and the uncertainty is estimated as the square root of the maximum number of anomalies. We consider the result significant if the difference is greater than the estimated uncertainty. A negative difference indicates that the number of anomalies detected after the earthquake is greater than the number of those detected before the earthquake. Normalization with respect to the total hours of geomagnetical quiet time is provided to take into account the possible imbalance in the investigated events due to our cut of the geomagnetic perturbed times (i.e., the ones when |Dst| > 20 nT and/or ap > 10 nT). The normalized quantities are provided in the second line of each cell with the label "Norm". Such imbalance due to geomagnetic activity can be due not only to sporadically geomagnetic storms but also to the different solar activity that was at a "minimum" at the beginning of the *Swarm* mission and is now in the increasing stage toward the next "maximum". The incomplete dataset in the first and last 500 days of analysis could enhance such a difference in the solar cycle. Most of the investigations have a significant differential number of anomalies that preceded the earthquake, supporting the hypothesis that some of these anomalies are induced by the preparation phase of earthquakes. The analysis with the higher percentage of pre-earthquake anomalies is the Y-East magnetic field in the period band of 2–10 s. Concerning the *Swarm Ne* anomalies in the period band of 25–50 s, we note a relatively high percentage considering the high number of total anomalies (40,219) compared with the result of the "Y, no-band". Furthermore, if we look at the general pattern, it is possible to suppose that the pre-seismic anomalies start about 400 days before the earthquake as the color of the concentration becomes darker (about orange, i.e., with a concentration of anomalies equal to or greater than seven anomalies/degree$^2$) until the earthquake and persists until about 250 days after the earthquake, when it seems to go back at a background level (which shows a yellow color corresponding to about 5.5 anomalies/degree$^2$). We recognize that such investigations are biased by the fact that higher magnitude earthquakes tend to show anomalies ahead of time, but such earthquakes are fewer in number (according to the Gutenberg–Richter magnitude distribution [42]), so most of the weight on the anomalies is posed by lower magnitude events. Even if 8 years of data seems like quite a long time, only a longer satellite dataset could help to better investigate this aspect. For this reason, we would like to underline the importance of maintaining the orbiting of the *Swarm* constellation (and other missions equipped with magnetometers, such as CSES) for as long as possible.

**Table 4.** Comparison of the number of anomalies in the closer bin to the epicenter before and after the earthquake for the symmetrical investigations (from −500 days to +500 days) of 8 years of *Swarm* data correlated with M5.5+ earthquakes by the WSC algorithm. A normalization ("Norm") is provided to take into account possible different geomagnetic activities occurring before and after each investigated event.

| Parameter, Period Band | Anomalies before the Earthquake | Anomalies after the Earthquake | Difference of the Anomalies | Estimated Uncertainty | Is the Result Significant? |
|---|---|---|---|---|---|
| Y, no-band | 16,630 Norm: 16,584 | 16,436 Norm: 16,481 | 194 (1.2%) Norm: 103 (0.6%) | 129 | No [1] |
| Y, 2–10 s | 4959 Norm: 4953 | 4563 Norm: 4568 | 396 (8.7%) Norm: 385 (8.4%) | 70 | Yes |
| Y, 10–25 s | 5078 Norm: 5066 | 4917 Norm: 4928 | 161 (3.3%) Norm: 137 (2.8%) | 71 | Yes |
| Y, 25–50 s | 6297 Norm: 6273 | 5858 Norm: 5880 | 439 (7.5%) Norm: 392 (6.7%) | 79 | Yes |
| *Ne*, no-band | 15,504 Norm: 15,469 | 15,618 Norm: 15,652 | −114 (−0.7%) −182 (−1.2%) | 125 | Yes, but post-seismic |

**Table 4.** *Cont.*

| Parameter, Period Band | Anomalies before the Earthquake | Anomalies after the Earthquake | Difference of the Anomalies | Estimated Uncertainty | Is the Result Significant? |
|---|---|---|---|---|---|
| *Ne*, 2–10 s | 3157 Norm: 3136 | 3337 Norm: 3360 | −180 (−5.4%) Norm: −224 (−6.7%) | 58 | Yes, but post-seismic |
| *Ne*, 10–25 s | 4661 Norm: 4669 | 4562 Norm: 4554 | 99 (2.2%) 114 (2.5%) | 68 | Yes |
| *Ne*, 25–50 s | 13,033 Norm: 12,991 | 12,538 Norm: 12,579 | 495 (3.9%) Norm: 413 (3.3%) | 114 | Yes |

[1] Such difference is not significant if we take into account the normalization, i.e., it could be due to unbalanced geomagnetic activity in the investigated period.

## 5. Conclusions

In this paper, we extended the work published by De Santis et al. [29] to a longer period (8 years) than that originally analyzed (4.7 years) of *Swarm* data correlated with M5.5+ shallow earthquakes. Here, we not only provide an analysis extended to the first 8 years of *Swarm* data but also introduce some new methodologies to improve the definition of the anomalies (by frequency analysis) and in terms of better investigation of the results—in particular, searching for possible influences of the seismo-tectonic context on the possible pre-earthquake electromagnetic signals. New methodologies and investigations are proposed to take into account the frequency of the signal and the different anticipation times of anomalies expected for several magnitude earthquakes. We found that several concentrations of ionospheric anomalies are statistically significant, well overpassing the given thresholds of $d \geq 1.5$ and $n \geq 4.0$—i.e., they are very much greater than the concentrations obtained from the space-time-homogenously-distributed anomalies, and we can infer that the earthquakes associated with such significant concentrations could be more likely "predicted", even if this is not the goal of the present work. By the ROC curve, we confirm that the anomalies selected by WSC and the aforementioned statistical criteria over "*d*" and "*n*" have predicting capabilities, showing an AUC that is 1.67 times better than that of a random predictor. Future studies could be oriented to filter the signals that are most prone to earthquake-induced anomalies (for example, Zhu et al. [50]) in order to better depict the signal "features" of possible seismo-induced phenomena in the ionosphere. Nevertheless, although such ionospheric pre-earthquake disturbances are still widely debated, there is some evidence, in the literature and in this paper, that they statistically exist.

Our main conclusions are:

1.  The anticipation time "$\Delta T$" of the anomaly increases with the magnitude of the incoming earthquakes following the Rikitake laws [37], with these specific coefficients being $\Delta T_{mag} = 10^{-0.96+0.51 \cdot M}$ and $\Delta T_{Ne} = 10^{-3.46+0.83 \cdot M}$ for the magnetic field "mag" and electron density data, respectively. The anticipation time of large earthquakes (M7.5+) seems to be some years before the event and has been detected.
2.  The focal mechanism seems to have a small or null influence on the generated frequency of the possible pre-earthquake anomalies.
3.  Earthquakes localized in the land areas tend to be preceded by lower frequency anomalous signals, while sea earthquakes are more likely to be preceded by faster signal anomalies.
4.  The *Swarm* magnetic field signal anomalies generally show a better correlation with earthquakes than the electron density ones do.
5.  A more selective set of parameters, achieved here by the investigation of the signal frequency, reduces the size of the anomaly dataset, and it is shown that the possible correlation with the seismic event has a higher statistical significance for both the magnetic field and *Ne* observations.
6.  Frequency analysis seems to be fundamental in some cases: for electron density, we find a higher correlation with anomalies, with a signal period in the range of 25–50 s.

7.  All the results in this paper have been tested with the "confusion matrix" approach, reaching an accuracy from 75% to 95% and an alarmed time-space from 0.7% to 19.1%. The real results show a predicting capability that is 1.67 times better than that of a random predictor, according to the AUC of the ROC curve, which further proves a prediction capability of the best detected ionospheric anomalies by WSC.

The future perspective of this work can include, but is not limited to, an improvement of the anomaly definition criteria and the extension to a larger dataset and to other satellites such as CSES-01. By the way, CNSA, together with the Italian Space Agency (ASI), is preparing a second CSES-02 satellite that is planned to be launched next year, forming the first-ever pair of satellites fully dedicated to searching for the possible ionospheric precursors of strong earthquakes.

Finally, we think that this study is fundamental to assessing the Lithosphere-Atmosphere-Ionosphere Coupling mechanism and for a future prediction system. Further, it integrates such kinds of studies with seismological investigations of the Earth's surface and chemical–physical analyses of the atmosphere (e.g., a preliminary comparison between magnetic anomalies and atmospheric surface temperature has been performed by Ghramy et al. [51]). So, as the earthquake is a complex phenomenon, by looking at all of the geo-layers together (as suggested in the "Geosystemics" approach by De Santis et al. [52]), it will be possible to better understand its physical features in its preparation phase and, hopefully, predict it one day.

**Supplementary Materials:** The following are available online at https://www.mdpi.com/article/10.3390/rs14112649/s1, Figure S1: Example of data processing to extract the anomaly, Figure S2: WSC applied to *Swarm* mag. from −500 to 500 days, Figure S3: WSC applied to *Swarm* mag. from −1000 days until the earthquake origin time, Figure S4: WSC applied to the shuffled magnitude earthquake catalog, Figure S5: Fit of anticipation time versus signal period, Table S1: Statistical evaluation of the results for the *Swarm* magnetic field, Figure S6: WSC applied to *Swarm Ne* with four different methods, Figure S7: WSC applied to *Swarm Ne* from −500 to 500 days, Figure S8: WSC applied to *Swarm Ne* in five earthquake magnitude bands, Table S2: Statistical evaluation of the results for the *Swarm Ne*, Figure S9: Comparison of the Rikitake law coefficients obtained in this work with the previous ones.

**Author Contributions:** Conceptualization, D.M. and A.D.S.; methodology, D.M., A.D.S. and S.A.C.; software, D.M., S.A.C. and M.S.; validation, S.A.C. and M.S.; formal analysis, D.M.; investigation, D.M.; resources, K.Z. and D.M.; data curation, D.M.; interpretation, D.M. and A.D.S.; writing—original draft preparation, D.M.; writing—review and editing, all authors; visualization, D.M.; supervision, A.D.S. and K.Z.; project administration, K.Z., D.M. and A.D.S.; funding acquisition, K.Z., D.M. and A.D.S. All authors have read and agreed to the published version of the manuscript.

**Funding:** This research was funded by the National Natural Science Foundation of China, grant number 41974084; the China Postdoctoral Science Foundation, grant number 2021M691190; the Italian Space Agency, grant number 2020-32-HH.0 (Limadou-Science+); the International Cooperation Project of the Department of Science and Technology of Jilin Province, grant number 20200801036GH; and the Italian Ministry of University and Research, grant number D53J19000170001 (Pianeta Dinamico—Working Earth).

**Data Availability Statement:** The *Swarm* satellite data are freely available from ftp and http servers (swarm-diss.eo.esa.int, last accessed on 3 January 2022). The USGS Earthquake catalog is freely accessible at https://earthquake.usgs.gov/earthquakes/map (accessed on 1 February 2022).

**Acknowledgments:** We warmly thank Francisco Javier Pavón-Carrasco for the software developed to analyze the *Swarm* data, and we would like to acknowledge Guido Ventura, Rita di Giovambattista, Loredana Perrone, Alessandro Piscini, Roger Haagmans, Giorgiana De Franceschi, Luca Spogli, Claudio Cesaroni, Ilaria Spassiani, and Matteo Taroni, who participated in some of the discussions about this work and its previous investigations carried out in the framework of the SAFE (SwArm For Earthquake study) ESA-funded project and in the following activities and projects.

**Conflicts of Interest:** The authors declare no conflict of interest. The funders had no role in the design of the study; in the collection, analyses, or interpretation of data; in the writing of the manuscript; or in the decision to publish the results.

## Appendix A. Evaluation of the Declustering of the Earthquake Catalog on the Worldwide Statistical Correlation Results

This appendix is dedicated to evaluating how the decluster technique that was used to pre-process the earthquake catalog could have affected the results.

Determining if an earthquake is an isolated event or part of a seismic swarm or sequence—and, in this last case, discriminating between foreshocks, mainshocks, and aftershocks—has been always a very challenging topic for seismologists [53]. Very recent frontiers of seismology are even trying to evaluate in real-time if an earthquake is a fore-shock or a mainshock by a traffic-light scheme based on the evaluation of the *b*-value of the Gutenberg–Richter distribution [54].

Declustering an earthquake catalog means the removal of the foreshocks and after-shocks in order to identify the "clusters" of seismicity led by the mainshock. The detection of the cluster is a difficult problem, and several approaches are available, such as that of Reasenberg [35] that we applied in this paper, which uses a radius around the event (a radius of 10 km for the present paper) and a time window to search for foreshocks (with 10 days in advance, for this paper) and aftershocks (in this paper: 20 days). If another earthquake is detected inside the space-time researched area, the same approach is applied until no other events are detected. The algorithm also takes into account the uncertainty of the earthquake localization that we selected from the values directly available in the USGS catalog. A different choice of such free parameters of the algorithm yields obviously different results. For example, in De Santis et al. [29], a shorter time window was used (1 day before and 10 days after), but we found that some events that occurred in the same tectonic context in a short time were not detected as the same cluster for such strict time windows; therefore, in the present work, we decided to use a slightly longer time window. As already presented in the discussion section, the statistical results are very consistent among the two works, demonstrating that, even with different decluster parameters in the Reasenberg approach and the slightly different earthquake catalog, the main results of our Worldwide Statistical Correlation (WSC) algorithm are not affected significantly. The reason for this is probably that, in the final WSC results, the bins are generally large in space (3.34°) and time (from 2.4 days to 20 days), so the low "space and time" resolution homogenizes the differences among the two declustered earthquake catalogs. Here, we want to explore how the results are affected by using another declustering technique—in particular, that provided by Urhammer [55]—or by using the original undeclustered catalog. More recent seismological techniques have been developed, such as the Epidemic-Type Aftershock Sequence (ETAS) model that was also applied to decluster earthquake cata-logs [56]. The ETAS technique permits the obtention of the probability that an earthquake can trigger another one (or be triggered by another one) and also has some forecast capabil-ities [57]. ETAS declustering has been applied to global seismicity by Nandan et al. [58], but this is beyond of the scope of this paper (to test it by WSC). We propose, as a future improvement, the study of the ionospheric electromagnetic disturbances combined with the ETAS investigation of seismicity. Different decluster techniques have been compared by Mizrahi et al. [59] in terms of the Gutenberg–Richter parameters obtained after decluster-ing the catalog and were also compared with the original catalog. From [59], it is possible to note that Reasenberg [35] and Urhammer [55] provide a different, non-superposed set of *a*- and *b*-values of the Gutenberg–Richter earthquake magnitude distribution, so we consider them as independent for the next test. We reanalyzed the *Swarm* magnetic field data presented in Figure 1 with the Urhammer declustered catalog and the earthquakes not declustered by associating all the anomalies to all the earthquakes (Method 1). In addition, we also propose Method 3 (maximum magnitude) to analyze the non-declustered catalog, as it could, in principle, provide similar results to a declustered catalog. In fact,

the declustering means extracting the mainshocks that have a higher magnitude than the foreshocks and aftershocks. So, associating the anomaly with the maximum magnitude earthquake could mean linking it with the mainshock. Contrariwise, we need to note that the seismic radius used by the decluster techniques tends to be very much smaller (e.g., 10 km) than the Dobrovolsky's radius used by our searching algorithm (e.g., ~600 km for M6.5), so it is not the same approach as that using the declustered catalog.

Figure A1 reports the results of the WSC applied to 8 years of *Swarm* magnetic field data and M5.5+ shallow earthquakes declustered by the above-mentioned techniques (on the left side: A and C) or not declustered (on the right side: B and D). In subfigure D, the method of "Max (magnitude)" has been applied to the WSC using the non-declustered earthquake catalog (i.e., the original one). It is possible to note that, in any case, some concentrations of anomalies at about 30 days and 80 days before the earthquake are underlined. Such evidence provides more proof that these concentrations are real and, furthermore, that they are not dependent on the way in which the earthquake catalog is treated. Comparing the two results with the declustered catalogs (subfigures A and C), it is possible to note that the results are very similar, even if the decluster technique is different. The pattern of concentrations is almost equal, with slightly differences, and the statistical significance is comparable. Even without declustering the earthquakes, some high-concentration bins are preserved, but such a choice leads to the overcounting of the anomalies with a value of 1.68 anomalies–EQ links over the number of anomalies associated with at least one earthquake. The overcounting is lower with the declustering by Reasenberg (1.36) [35] compared to that obtained by applying Urhammer's technique (1.41) [55], so even if the statistical significance in terms of $d$ and $n$ is a bit lower with Reasenberg [35], we think that such a technique is a better choice than that of Urhammer [55], as it has a lower overcounting of anomalies. Finally, the choice of the maximum magnitude with the non-declustered earthquake catalog confirms the other results; however, it also loses some seismic information that the declustering algorithm takes into account. In fact, this technique operates on a distance that, in several cases, goes behind the typical stress-interaction region between the earthquakes, as already mentioned above.

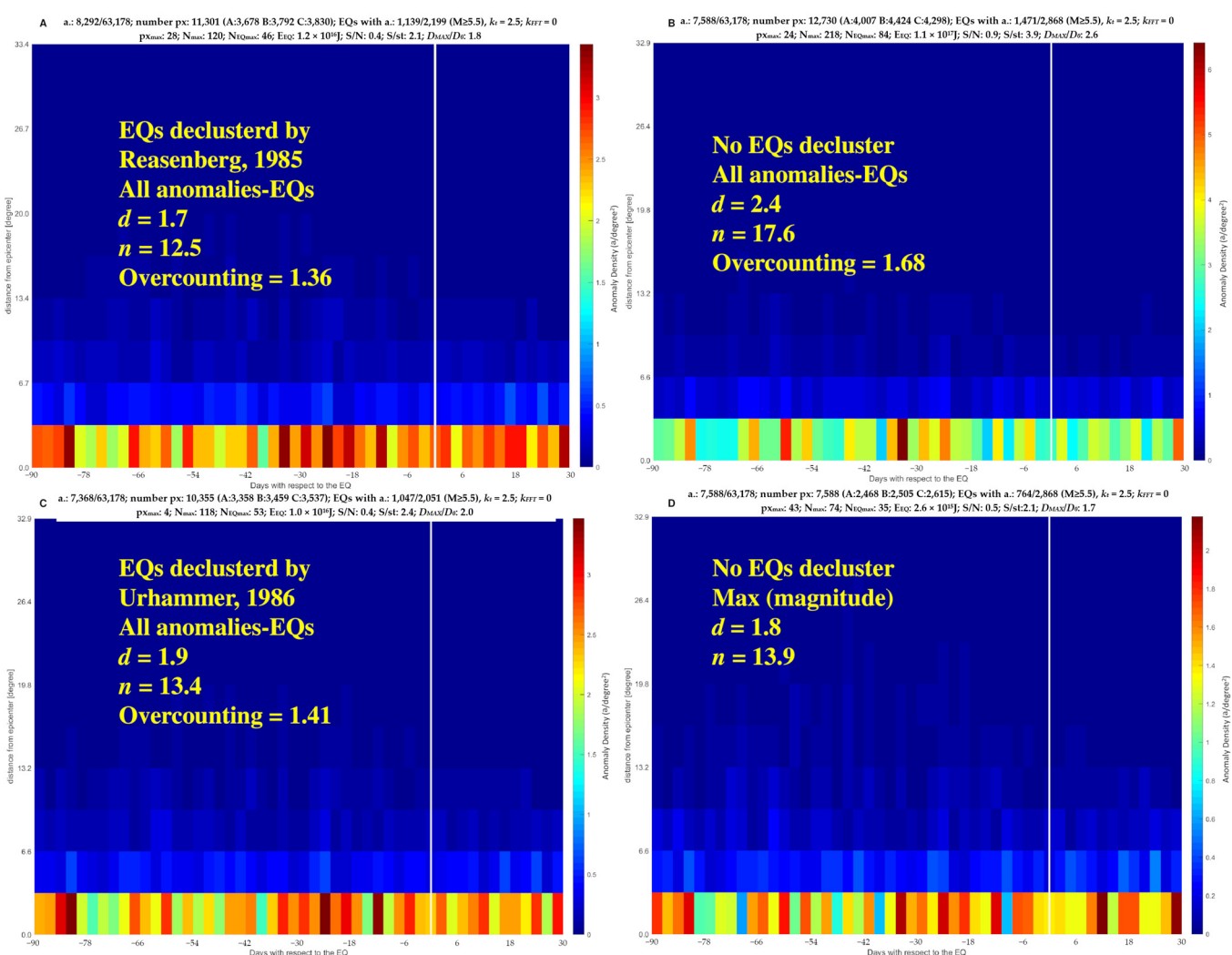

**Figure A1.** Comparison of the results of the Worldwide Statistical Correlation using two different declustering methods or by not declustering the catalog. The *Swarm* magnetic field data investigated from 90 days before the earthquake until 30 days after it are used for this example. The earthquake catalog has been declustered with: (**A**) Reasenberg, 1985 [35]; (**C**) Urhammer, 1986 [55]. The earthquake catalog used for the WSC of (**B**,**D**) has not been declustered. In (**A–C**), all anomalies have been assigned to any compatible earthquake, and the overcounting factor (the number of links between the anomalies and EQs divided by the number of anomalies with EQs) has been reported, while, in (**D**), each anomaly has been connected only to the higher magnitude earthquake among the possible ones (thus, there is no overcounting).

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
