# Peer review of "Worldwide Statistical Correlation of Eight Years of Swarm Satellite Data with M5.5+ Earthquakes: New Hints about the Preseismic Phenomena from Space"

_remotesensing, doi:10.3390/rs14112649_

Round 1

Reviewer 1 Report

I have no comment for this article.

Author Response

Dear reviewer,

We thank you so much for your consideration and evaluation of our manuscript.

Reviewer 2 Report

Manuscript Number: remotesensing-1738157

Full Title: Worldwide Statistical Correlation of eight years of Swarm satellite data with M5.5+ earthquakes: new hints about the preseismic phenomena from space.

The paper submitted to Remote Sensing MDPI by D. Marchetti et al.  perform a new methodologies applied a Worldwide Statistical Correlation approach to earthquakes using the dataset of Swarm magnetic field and electron density signals and provide some new results on how earthquake features could influence ionospheric electromagnetic disturbances.

The paper is analysed in revised form.

The topic is of considerable interest to the scientific community, particularly because the implications between earthquakes and magnetic field and electron density signals is of great scientific interest because of the potential impact on many disciplines.

The paper has been edited according to the RS instructions for authors and is set up correctly in all its paragraphs.

In my opinion, the study is adequately original and the methodological approach is rigorous. The introduction synthesizes the scientific background. The methods and tools are described with details to allow another researcher to reproduce the results The results are interpreted in appropriate way and the data are robust enough to draw the conclusions.
There are only a few minor comments that the authors should reply before being published, concerning the readability of figures 1 to 9. In fact, the labels within them are not readable, so I ask the authors to amend them for the readers. The figures themselves are in low quality and this is not possible in a prestigious journal like RS.

In conclusion the work deserves publication after minor revisions, in my opinion.

Best regards

Author Response

Dear reviewer,

We thank you so much for your time, consideration and detailed and positive evaluation of our manuscript. We are delighted that the reviewer found our paper and research of interest and well described.

For what concerns the figure we are very sorry for the low-quality version in the pdf for revision. We have revised all the figures and assure the quality is good for the final version. Furthermore, we replace the labels by increasing font size when it was necessary.

Thank you again for your suggestions and consideration of our manuscript.

Reviewer 3 Report

Please see attached pdf-File.

Author Response

Dear reviewer,

Thank you so much for your precious indications.

Please see the attachment for a reply point-by-point to your precious comments and suggestions.

Reviewer 4 Report

Accept without corrections

Author Response

Dear reviewer,

We thank you so much for your consideration and positive evaluation of our manuscript.

This manuscript is a resubmission of an earlier submission. The following is a list of the peer review reports and author responses from that submission.

Round 1

Reviewer 1 Report

See my attached document.

Author Response

Dear gentle reviewer,

Thank you so much for your high valuable indications,

Please find in attach our reply point-by-point to your questions and valuable suggestions,

Thank you a lot! 

Reviewer 2 Report

  1. Authors should fix a few technical issues. The phrase “Error! Reference not found.” appears multiple times across the manuscript.

  1. Authors should use more strict and explicit formulations. Statements like “a bit more probable” (see line 647) or “to be less likely” (line 649) may confuse a reader. Do the authors express the uncertainty of results being reported?

  1. On lines 679-683 the authors describe electron density changes in terms of colors ( “becomes darker (about orange)”). It is better to provide numerical values (and units) that correspond to those changes.

  1. Along the manuscript, the authors denote earthquakes’ magnitudes as M. Which specific M do they use? There are at least Mw, Ms, Ml, and Mk in use (others are skipped here).

  1. In the introduction, the authors list just a few possible mechanisms of seismo-ionosphere coupling. A more extensive review is available in [Liperovsky et al., 2008, https://doi.org/10.1134/S0016793208060133 ].

  1. In the introduction, the authors refer to Kuo et al [2014,2018] simulation results. These are an important attempt to quantify seismo-ionosphere coupling processes which, however, has an open topic for discussion. Denisenko et al. [2018a, https://doi.org/10.1016/j.jastp.2018.09.002 ] criticized a few models like in [Kuo et al., 2014], and investigated it explicitly in [Denisenko et al., 2018b, https://doi.org/10.1029/2018JA025228 ].

  1. There are a large number of publications that analyzed ionosphere satellite electric fields observations over seismically active regions. The majority of observations were obtained from INTERCOSMOS-BULGARIA-1300 and DEMETER satellites (see a review in [Zolotov, 2015, https://doi.org/10.1134/S1990793115050255 ]). In this paper, the authors evaluate magnetic field disturbances. Are there any benefits in analyzing magnetic field components? Do CSES and Swarm provide electric field data? Is the method specifically designed for magnetic fields measurements?

  1. Authors introduce some new ‘d’ and ‘n’ coefficients. Can those coefficients be interpreted in terms of ‘prediction’ of a forthcoming earthquake?

  1. Authors should explicitly provide the time (number of days) when they expect to find precursors for the forthcoming earthquake. How do those values contrast with other researchers' results?

  1. For CSES data, the authors used another k-value to determine the anomaly threshold (see line 404). Does it influence the statistical significance of the results? Are there any considerations on k-values to keep the results reliable (in a statistical sense)?

  1. In conclusion (lines 703-704), the authors state that the anticipation time of large earthquakes (M7.5+) seems to be some years before the event. Did they perform similar estimations of the anticipation time for earthquakes with smaller magnitudes? If yes, does it agree with other publications?

Author Response

Dear reviewer,

We would thank you so much for your precious suggestions and good question that helped us to improve the manuscript and describe better our results. We provide in bold black a specific answer to the valuable questions that you pointed out and what we have changed or improved to address your good observations:

  1. Authors should fix a few technical issues. The phrase “Error! Reference not found.” appears multiple times across the manuscript.
  • We are very sorry for the mistake in the references and we checked carefully them in the revised version.
  1. Authors should use more strict and explicit formulations. Statements like “a bit more probable” (see line 647) or “to be less likely” (line 649) may confuse a reader. Do the authors express the uncertainty of results being reported?
    • Thank you very much for your question and suggestion. We have inserted all the percentages close to the expressions “a bit more probable”, “to be less likely” and also in the other part of the sentences to provide also in the text the values of deviation from the standard distribution of earthquakes. For the focal mechanism we consider significative a deviation that is greater than 5% and we used the bold in the table 1 to underline such cases. We add this indication also in the caption of the table. In the case of sea/land earthquake we consider also that the percentages are monotonic i function of the frequency and we think this is another evidence of their link as stated in the text.
  1. On lines 679-683 the authors describe electron density changes in terms of colors ( “becomes darker (about orange)”). It is better to provide numerical values (and units) that correspond to those changes.
    • We thank he reviewer for his suggestion! We have added the numerical indications that correspond to the described color providing the values both for the higher concentration (>7 anomalies/degree2), as well as for the identified background (5.5 anomalies/degree2).
  1. Along the manuscript, the authors denote earthquakes’ magnitudes as M. Which specific M do they use? There are at least Mw, Ms, Ml, and Mk in use (others are skipped here).
    • Thank you very much for your very important question. Yes the reviewer is right that several magnitudes exist. We got our earthquake catalogue from USGS where they provided several magnitude types, among them generally the Mw (moment magnitude) is the preferred. If Mw was not available the Mb (duration) is instead provided). We insert in the data description a specific sentence to better specify this important point:
      “from the USGS global catalogue that provides an estimation of the hypocenter, origin time and magnitude (generally the moment magnitude Mw calculated from centroid moment tensor inversion, if not available the duration magnitude mb or other).”
  2. In the introduction, the authors list just a few possible mechanisms of seismo-ionosphere coupling. A more extensive review is available in [Liperovsky et al., 2008, https://doi.org/10.1134/S0016793208060133 ].
    • Thank you so much for your suggestion of a such wonderful review paper about several LAIC models. We have included in the Introduction as suggested and also in the Discussion.
  1. In the introduction, the authors refer to Kuo et al [2014,2018] simulation results. These are an important attempt to quantify seismo-ionosphere coupling processes which, however, has an open topic for discussion. Denisenko et al. [2018a, https://doi.org/10.1016/j.jastp.2018.09.002 ] criticized a few models like in [Kuo et al., 2014], and investigated it explicitly in [Denisenko et al., 2018b, https://doi.org/10.1029/2018JA025228 ].
  • Thank you for the suggested papers. We integrated them in the introduction and we also ad other two references that were a reply from Prokhorov and Zolotov and the reply to reply of Kuo and Lee:
    “Such simulation [i.e. Kuo et al., 2014, 2018] opened a scientific discussion, in fact, Prokhorov and Zolotov (2007) found that some assumptions in the model could be too simplified, and Kuo and Lee (2017) replied with an update of the original work showing that the model seems anyway valid. Furthermore, Denisenko et al., (2018a, b) found some possible inaccuracies in the model of Kuo and they proposed another model also improving the calculus of the conductivity in the ionosphere but the value on the ground of electric field signal necessary to create a small perturbation of the ionosphere was found very large, even if not impossible. As stated by the same authors of all these papers more pieces of evidence are necessary to understand if such phenomena exist and which could be the more reliable model to describe them. The purpose of this paper is to provide some observational points for such discussion.”
    References:
    • Prokhorov, B. E., and Zolotov, O. V. (2017), Comments on “An improved coupling model for the lithosphere-atmosphere-ionosphere system” by Kuo et al. [2014], J. Geophys. Res. Space Physics, 122, 4865– 4868, doi:10.1002/2016JA023441.
    • Kuo, C.-L., and Lee, L.-C. (2017), Reply to comment by B. E. Prokhorov and O. V. Zolotov on “An improved coupling model for the lithosphere-atmosphere-ionosphere system”, J. Geophys. Res. Space Physics, 122, 4869– 4874, doi:10.1002/2016JA023579.
    • Denisenko, V. V., Boudjada, M. Y., & Lammer, H. (2018a). Propagation of seismogenic electric currents through the Earth's atmosphere. Journal of Geophysical Research: Space Physics, 123, 4290– 4297. https://doi.org/10.1029/2018JA025228
    • Denisenko, V.V., Nesterov, S.A., Boudjada, M.Y., Lammer, H. (2018b) A mathematical model of quasistationary electric field penetration from ground to the ionosphere with inclined magnetic field, Journal of Atmospheric and Solar-Terrestrial Physics, 179, 2018, 527-537, https://doi.org/10.1016/j.jastp.2018.09.002.

  1. There are a large number of publications that analyzed ionosphere satellite electric fields observations over seismically active regions. The majority of observations were obtained from INTERCOSMOS-BULGARIA-1300 and DEMETER satellites (see a review in [Zolotov, 2015, https://doi.org/10.1134/S1990793115050255 ]). In this paper, the authors evaluate magnetic field disturbances. Are there any benefits in analyzing magnetic field components? Do CSES and Swarm provide electric field data? Is the method specifically designed for magnetic fields measurements?
  • We thank a lot the reviewer for such a suggested paper that we integrated and discussed in the introduction and we also inserted some other works references to provide a more complete background literature:
    “Previous statistical works have provided pieces of evidence in the correlation of the ionospheric disturbances in terms of electron density measured by DEMETER satellite founding a significant increase from 10 to 6 days prior to the seismic M4.8+ events [24, 25] and also increase of electric field (typically around 10/20 mV/m) from few minutes to some days prior to several M5+ reported by Zolotov [26] using DEMETER and INTERCOSMOS-BULGARIA-1300 satellites.”
    1. Parrot M., Li, M., 2015. "Demeter results related to seismic activity," in URSI Radio Science Bulletin, vol. 2015, no. 355, pp. 18-25, Dec. 2015, doi: 10.23919/URSIRSB.2015.7909470.
    2. Yan, R., Parrot, M. & Pinçon, J.-L. (2017). Statistical study on variations of the ionospheric ion density observed by DEMETER and related to seismic activities. Journal of Geophysical Research: Space Physics, 122, 12,421–12,429. https://doi.org/10.1002/2017JA024623
    3. Zolotov, O.V. Ionosphere quasistatic electric fields disturbances over seismically active regions as inferred from satellite-based observations: A review. Russ. J. Phys. Chem. B 9, 785–788 (2015). https://doi.org/10.1134/S1990793115050255
  • Empirically we found that magnetic field can provides better statistical reliability with respect to the electron density (De Santis et al., 2019 and this paper) but we never compared with electric field disturbances.
  • Unfortunately, Swarm is not provided with an instrument to measure the Electric field Vector, it can only measure the spacecraft electric potential, the electron density by Langmuir Probes and the Faceplate detector of EFI. On the other hand, CSES-01 is provided with Electric Field Detector (EFI) instrument. The instrument placed in the body of the satellite uses 4 sensors placed on a long (about 4 meters) boom and it measures the Electric field vector. As stated in one of the following points the CSES data investigation has been removed to dedicate a future deeper paper on it and the investigation of CSES EFD data could be included.
  • Yes, our method has been specially designed for magnetic data analysis developing an algorithm that calculates and analyses the magnetic residuals track-by-track (MASS, described in the paper). We extended the algorithm to electron density data investigation, considering the difference of the original measurements (for example the data sampling rate is 2 Hz for Ne instead of 1Hz for the Swarm LR MAG product data).
  1. Authors introduce some new ‘d’ and ‘n’ coefficients. Can those coefficients be interpreted in terms of ‘prediction’ of a forthcoming earthquake?
  • Thank you for such an intriguing and good question. Basically, “d” and “n” represent how much the concentration of anomalies is greater than space-time homogeneous random one, so they don’t are related o predictability. Anyway if we consider that the earthquakes contained in maximum concentration (their number is reported in the second line of the figure title as NEQmax) are more likely to be predicted when d and n are large, for example, we already defined as significant the results with d ≥ 1.5 and n ≥ 4.0 (De Santis et al., 2019) and the earthquake preceded by anomalies with such characteristics are more likely to be predicted, so we could even say as the reviewer suggested that higher d and n provide great predictability of the associated earthquakes. We inserted a specific sentence in the conclusions:
    “We found that several concentrations of ionospheric anomalies are statistically significant well overpassing the given thresholds of d ≥ 1.5 and n ≥ 4.0, i.e., they are very much greater than concentrations obtained from space-time homogenously distributed anomalies and we can infer that the earthquakes associated with such significant con-centration could be more likely “predicted”, even if this is not the goal of the present work.”
  1. Authors should explicitly provide the time (number of days) when they expect to find precursors for the forthcoming earthquake. How do those values contrast with other researchers' results?
  • Thank you very much for your question. Firstly we would underline that at present we are not developing a prediction method, but we are analysing in a post-event approach if the ionosphere was perturbed in the days, months or years before the upcoming earthquakes. In this paper, we used a time-window of 90 days, 500 days, and 1000 days before the earthquake. Anyway, we would like to focus the attention of the reviewer on an interesting law that we further confirm in this paper that was proposed by Rikitake and it predicted that the logarithm of anticipation time is directly proportional to the magnitude of the incoming earthquake. For magnetic data, we calculated (see figure 2a) how much is the anticipation time and it is: 10-4.07+0.93M (days), as we analysed earthquake in magnitude range from 5.5 to 8.3 this corresponds to a time from 11 days for M5.5 to a little less than 10 years for M8.2. Even if this means that there is not a unique answer to your question, for a specific magnitude there is a precise answer, for example for M6.5 the predicted anticipation time of magnetic anomalies is 94 days. We inserted the anticipation time as Rikitake-law in the first point of the conclusion that we modified in the following way:
    “The anticipation time “∆aT” of the anomaly increases with the magnitude of the incoming earthquakes following the Rikitake-laws with these specific coefficients: and for magnetic “mag” and electron density data. The anticipation time of large earthquakes (M7.5+) seems to be some years before the event and has been detected.”
  1. For CSES data, the authors used another k-value to determine the anomaly threshold (see line 404). Does it influence the statistical significance of the results? Are there any considerations on k-values to keep the results reliable (in a statistical sense)?
    • We are sorry, but we decided to remove the CSES-01 analyses from the present paper to dedicate a specific paper on this analysis with further investigations. Anyway, to reply to your question, you are fully right the kt threshold used to extract the anomalies was different for Swarm and CSES. In principle we can expect that this threshold is independent of the investigated satellite, but as kt is a threshold over the root mean square of geomagnetic quiet tracks it is necessary to adjust it in the function of how much the signals are noisy. Normally a kt equal or greater than 1.5 or 2.0 (if not associated to kFFT) provide a reliable statistical analysis that in case the residuals are normally distributed it corresponds to 1.5 or 2.0 standard deviations. We will put more details on kt between Swarm and CSES in the future dedicated paper with the results of CSES-01.
  1. In conclusion (lines 703-704), the authors state that the anticipation time of large earthquakes (M7.5+) seems to be some years before the event. Did they perform similar estimations of the anticipation time for earthquakes with smaller magnitudes? If yes, does it agree with other publications?
    • Thank you for your great question. Yes, sure we have estimated the anticipation time for the whole investigated earthquake (M5.5+) but found better results for M6+ in the above-mentioned Rikitake law. We have reformulated this Conclusion point (as already stated). We underlined before only the M7.5+ because the concentration of anomalies was not found in the previous paper De Santis et al., Scientific Report, 2019 due to the too “short” analysed time (4.7 years) instead of the present 8 years of data. The anticipation times, in terms of Rikitake law, have been compared not only with De Santis et al., 2019, but also with a very early work of Rikitake, 1987 (see Figure S7 - old S9- of supplementary materials). Even if the work of 1987 was based only on ground data the coefficients that we found are very in agreement with such work (last paragraph of the Discussion section 4.1).

Reviewer 3 Report

The title “Worldwide statistical correlation of satellite data (Swarm and CSES) with M5.5+ earthquake: new hints about the preseismic phenomena” is an interesting paper. In my understanding, it is generally indispensable in discussing prediction problems to consider performance matrix such as true positives, true negatives, false positives and false negatives. This study, however, discussed only from the true positives (anomalies before and after truly happened earthquakes). Although the reviewer has imagined that many magnetic anomalies can be found even if earthquakes did not happen, the authors did not discuss such false predictions. That’s why the reviewer could not judge the methodology and the results obtained in this study were justified. Furthermore, the results in this study seemed just an extension of the previous work by De Santis et al. (2019), and I could not find any significant improvement from the previous methods in this paper. The references links were missing after L192. From the scientific reasons shown above, I am very sorry to reject the paper.

Author Response

(The authors gave the same response as above.)

Round 2

Reviewer 1 Report

I read the cover letter sent by the authors and appreciate the efforts they did to improve the manuscript.   I have anyway one last requirement which concerns the correction of the introduction. In the latter, they follow my suggestion to mention Freund's theory about p-holes. In the same paragraph, they claim that "he proposed that the detected Thermal-Infrared anomalies [e.g., 10] could be explained by Joule heating induced by the current generated by the p-holes". First, this is misleading because reference [10] doesn't feature Freund as a co-author. So this interpretation is not his. Indeed, this interpretation betrays Freund's spirit.
Freund's theory, as explained in the reference I provided in my review, claims that peroxy bonds
are reconstructed at the Earth surface, and this recombination emits photons in the same wavelength range covered by Thermal Infra Red detection - but there is no heating at all, thus no real temperature change. Just an anomaly due to some extra-photons that happen to have a similar wavelength.   Beyond this, the paper has been improved, yet the flow of English and writing is still not top-notch. For this latter aspect, I thus leave to the Editor the decision to publish it as it is, or to require a more polished version.

Reviewer 3 Report

I would appreciate the authors to honestly add the confusion matrices for discussing the accuracies. But I have further questions for the confusion matrices as below.

L550-L551

Although the authors described “A prediction can be considered good if A<50%”, I do not agree this. It must depend on the number of the classified samples. For example, when TP=100, FP=0, FN=0 and TN=10, AT can be higher than 50 % (AT=90%) despite the good prediction.

L553-574:

As partially described by the authors, the overall accuracies could be high when the number of True negatives is largely dominant. For example, when all the data are classified to “No anomalies”, the accuracy can be higher than 90% despite nothing was detected. It means that the detection accuracy must be discussed using Hit Rate and False Rate. According to the confusion matrices in Table 1, HR and FR were very low. Although the authors discussed in this section, I have to conclude that the magnetic and electron density data cannot be used as indicators of preseismic phenomena.

L553:

The reference link was also missing.

Table 1:

Some words in the left columns are missing (“90 days before the…”).